# Efficient Audio-Visual Speech Separation with Discrete Lip Semantics and Multi-Scale Global-Local Attention

**Kai Li**[1,2,*]**, Kejun Gao**[1,*]**& Xiaolin Hu**[1,2,3,†]

1. Department of Computer Science and Technology, Institute for AI,
BNRist, Tsinghua University, Beijing 100084, China
2. IDG/McGovern Institute for Brain Research, Tsinghua University, Beijing 100084, China
3. Chinese Institute for Brain Research (CIBR), Beijing 100010, China
`{li-k24, gkj23}@mails.tsinghua.edu.cn`
`xlhu@tsinghua.edu.cn`

## Abstract

Audio-visual speech separation (AVSS) methods leverage visual cues to extract target speech and have demonstrated strong separation quality in noisy acoustic environments. However, these methods usually involve a large number of parameters and require high computational cost, which is unacceptable in many applications where speech separation serves as only a preprocessing step for further speech processing. To address this issue, we propose an efficient AVSS method, named *Dolphin*. For visual feature extraction, we develop *DP-LipCoder*, a dual-path lightweight video encoder that transforms lip-motion into discrete audio-aligned semantic tokens. For audio separation, we construct a lightweight encoder–decoder separator, in which each layer incorporates a global–local attention (GLA) block to efficiently capture multi-scale dependencies. Experiments on three benchmark datasets showed that Dolphin not only surpassed the current state-of-the-art (SOTA) model in separation quality but also achieved remarkable improvements in efficiency: over 50% fewer parameters, more than $2.4\times$ reduction in MACs, and over $6\times$ faster GPU inference speed. These results indicate that *Dolphin* offers a practical and deployable solution for high-performance AVSS in real-world scenarios. Our code and demo page are publicly available at this link.

## 1 Introduction

In real-world environments, target speech is often masked by background noise and competing talkers. This challenge relates to the well-studied "cocktail party effect", where humans can selectively attend to a single speech in a noisy mixture (Cherry, 1953). Such observations have motivated extensive research on speech separation. However, audio-only speech separation methods frequently exhibit degraded performance in complex acoustic environments. By contrast, the integration of synchronous visual cues offers enhanced robustness to noise and has thus motivated extensive research into AVSS methods (Michelsanti et al., 2021).

In recent years, the application of deep learning methods to AVSS has attracted considerable attention (Gao & Grauman, 2021; Pegg et al., 2023; Wu et al., 2019; Martel et al., 2023; Li et al., 2024b; Zhao et al., 2025). Although many methods focus on performance, they often result in computationally intensive models (such as AV-Mossformer2 (Zhao et al., 2025)) that are not suitable for practical deployment. Alternative methods like RTFSNet (Pegg et al., 2023) and AVLiT (Martel et al., 2023) aim to improve efficiency using lightweight iterative separators, but the iterative process itself also incurs significant computational overhead. Thus, striking an appropriate balance between separation quality and computational cost remains a major challenge in AVSS.

---

[*]Kai Li and Kejun Gao contribute equally to the article.
[†]Corresponding author.

Apart from the computational cost of the audio separator, in the AVSS task, a long-standing and even more critical challenge arises from the "path dependence" on video encoders (Martel et al., 2023). Most methods employ large-scale visual backbone networks that are pre-trained on the lip reading task (Gao & Grauman, 2021; Pegg et al., 2023; Wu et al., 2019; Li et al., 2024a;b; Zhao et al., 2025). These backbone networks employ a larger number of parameters to extract semantically aligned features from lip movements, resulting in improved speech separation performance. However, this also entails an exceptionally high computational cost for high-performance separation systems. Previous studies have been caught in a dilemma: on the one hand, directly compressing existing large-scale visual encoders often leads to a significant loss in semantic representation capability and a drastic drop in separation performance (Wu et al., 2023); on the other hand, designing lightweight encoders from scratch, tailored for low-level tasks such as video reconstruction, tends to yield only shallow, pixel-level features and fails to extract effective semantic information, resulting in suboptimal separation results (Martel et al., 2023). Therefore, designing video encoders that are both lightweight and capable of sufficient semantic alignment remains a major challenge in AVSS research.

To address this issue, we propose a novel and efficient AVSS model, named *Dolphin*, which aims to improve computational efficiency and separation accuracy. We designed a lightweight dual-path video encoder with vector quantization (VQ) named DP-LipCoder to reduce computational cost while maintaining representational capacity. This video encoder maps the video frames into two visual features: one corresponding to compressed abstract visual features that preserve spatio-temporal structures, and the other corresponding to audio-aligned visual features. Specifically, by incorporating VQ (Esser et al., 2021) and leveraging knowledge distillation from a pretrained audio-visual (AV) representation model AV-HuBERT (Shi et al., 2022), we map continuous video streams into discrete semantic tokens that are highly aligned with audio.

In addition, we also introduce a lightweight separator with a single-iteration, global–local collaborative design. The separator is built upon TDANet (Li et al., 2023), which leverages a top–down attention mechanism. Unlike prior methods that rely on multiple iterations, our approach retains only one forward pass of the separator, while introducing global–local attention (GLA) blocks at each layer to mitigate performance loss. The global attention (GA) block employs coarse-grained self-attention (CSA) to capture long-range dependencies in a low resolution space, whereas the local attention (LA) block adopts heat diffusion attention (HDA), derived from the heat diffusion equation (Cannon, 1984), to smooth features across channels, suppress noise, and preserve details. With this design, we aim to achieve a balance between computational cost and separation quality.

We validated the effectiveness of Dolphin on three public AVSS benchmark datasets (Afouras et al., 2018a;c; Chung et al., 2018). Experiments showed that compared with the SOTA method IIANet, Dolphin performed better on all separation metrics, while also having significant advantages in resource efficiency: more than 50% reduction in parameters, over $2.4\times$ reduction in computational cost, and more than $6\times$ improvement in GPU inference speed. These results indicate that Dolphin offers a viable solution for AVSS on edge devices.

## 2 RELATED WORKS

### 2.1 AUDIO-VISUAL SPEECH SEPARATION

AVSS methods leverage visual cues like lip movements for robust speech separation in a noisy environment. Time–frequency (TF) domain methods transform speech signals into the time–frequency representation using the Short-Time Fourier Transform (STFT) for processing (Afouras et al., 2018b;a; Gao & Grauman, 2021). End-to-end time-domain models operate directly on raw audio waveforms, a direction first pioneered by AV-ConvTasNet (Wu et al., 2019). Following this line of research, a series of more advanced SOTA architectures such as CTCNet (Li et al., 2024a), IIANet (Li et al., 2024b) and AV-Mossformer2 (Zhao et al., 2025) have been proposed. Despite their improved separation quality, these models are computationally expensive, which limits their applicability in resource-constrained or latency-sensitive scenarios. While lightweight iterative separator reduce parameters, they suffer from inference latency due to their recurrent computation (Pegg et al., 2023; Martel et al., 2023; Sang et al., 2025). In contrast, we propose a single-iteration encoder–decoder architecture that, by modeling multi-scale global and local features within each layer, achieves a better balance between computational cost and performance.

## 2.2 PRETRAINED VIDEO ENCODER

In the AVSS task, the video encoder plays a decisive role in determining overall performance. Existing work typically faces a dilemma: On one hand, to extract visual features that are well aligned with speech semantics, researchers predominantly rely on large-scale backbone networks pretrained on lip-reading tasks (Gao & Grauman, 2021; Pegg et al., 2023; Wu et al., 2019; Li et al., 2024a;b; Zhao et al., 2025). While this strategy leads to improved performance, the substantial computational cost severely limits the model's practical applicability. On the other hand, efforts to reduce this cost have not yielded satisfactory results. Methods such as directly compressing these large models or designing lightweight networks from scratch for low-level reconstruction tasks face a common problem: a marked decline in semantic representation capability, resulting in poor separation performance (Wu et al., 2023; Martel et al., 2023). This situation appears to enforce a trade-off rule in the field: performance and efficiency cannot be achieved simultaneously. To solve this issue, we propose a novel video encoder. By jointly optimizing semantic extraction and video reconstruction, our method maintains lightweight characteristics while generating features with higher semantic density, thus providing stronger visual cues for speech separation.

## 3 METHODS

### 3.1 OVERALL PIPELINE

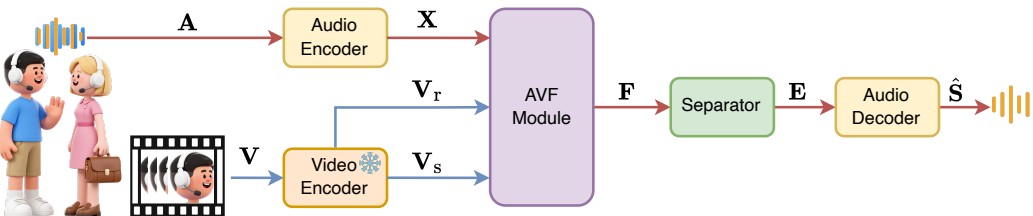

Figure 1: The overall pipeline of Dolphin. Frozen parameters are displayed with a snowflake marker.

Let $\mathbf{S} \in \mathbb{R}^{1 \times L_a}$ and $\mathbf{V} \in \mathbb{R}^{H \times W \times T_v}$ represent the input audio and video streams of the target speaker, where $L_a, T_v$ are the sequence length of the input audio and video separately, and $H$ and $W$ represent the height and width of the video frame. Given a mixture audio $\mathbf{A} \in \mathbb{R}^{1 \times L_a}$ containing utterances of target speaker and $C$ other speakers $\mathbf{B}_i \in \mathbb{R}^{1 \times L_a}$, along with noise $\mathbf{n} \in \mathbb{R}^{1 \times L_a}$:

$$\mathbf{A} = \mathbf{S} + \sum_{i=1}^{C} \mathbf{B}_i + \mathbf{n}. \tag{1}$$

The AVSS task is to recover $\mathbf{S}$ from $\mathbf{A}$ with the assistance of the video cues of target speaker $\mathbf{V}$. As shown in Figure 1, the Dolphin consists of five primary components: a pretrained video encoder, an audio encoder, an audio-visual fusion (AVF) module, a separator, and an audio decoder.

Specifically, the visual stream $\mathbf{V}$ is first fed into a pretrained video encoder. The resulting feature maps are then flattened to extract two types of features: reconstruction-related features $\mathbf{V}_r \in \mathbb{R}^{N_v \times T_v}$ and semantics-related features $\mathbf{V}_s \in \mathbb{R}^{N_v \times T_v}$. Here, $N_v$ is the dimension of the flattened feature vector, obtained by merging the channel dimension with the spatial dimensions $(h, w) = (\frac{H}{2^D}, \frac{W}{2^D})$, where $D$ represents the number of downsampling. $T_v$ denotes the number of temporal frames. For the audio stream, we employ a 1D convolutional layer as the audio encoder to encode the mixture audio $\mathbf{A}$, yielding the audio features $\mathbf{X} \in \mathbb{R}^{N_a \times T_a}$, where $N_a$ denotes the number of audio feature channels and $T_a$ denotes the length of audio feature. The visual features $\mathbf{V}_r, \mathbf{V}_s$ together with the audio features $\mathbf{X}$ are then fed into the AVF module, producing fused features $\mathbf{F} \in \mathbb{R}^{N_a \times T_a}$. Then, we feed $\mathbf{F}$ into the separator to obtain the target speaker's features $\mathbf{E} \in \mathbb{R}^{N_a \times T_a}$. Finally, the audio decoder employs one 1D transposed convolutional layer to transform $\mathbf{E}$ into the time-domain signal $\hat{\mathbf{S}} \in \mathbb{R}^{1 \times L_a}$, which corresponds to the separated speech of the target speaker.

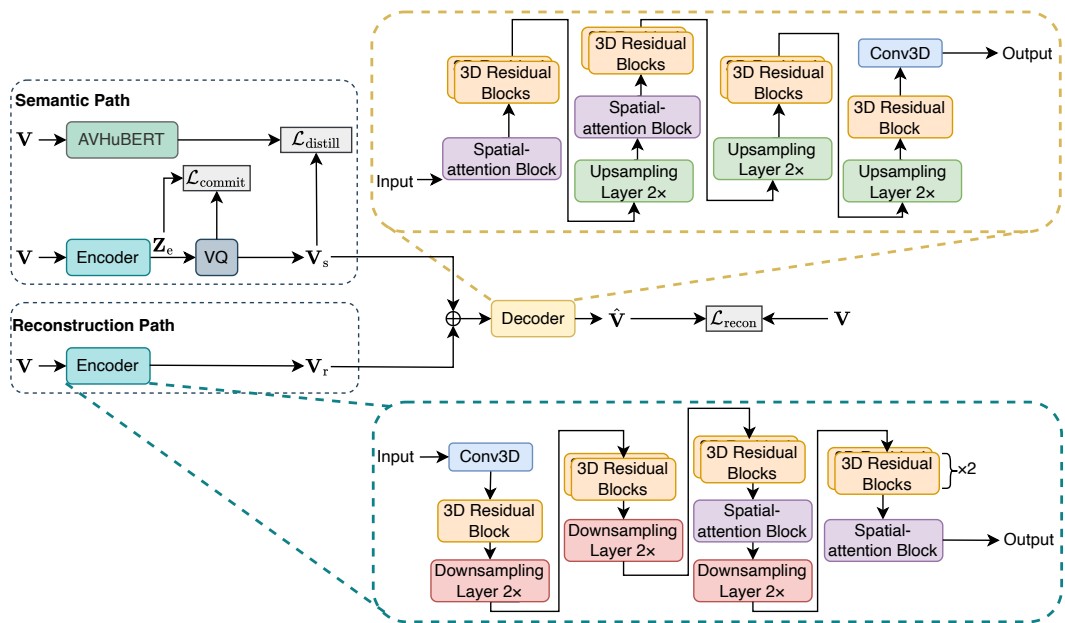

Figure 2: Overall pipeline of the AVDP-MagVIT network. The stacked 3D residual blocks denote two 3D residual blocks connected sequentially. The three loss functions are shown as gray blocks.

## 3.2 PRETRAINED VIDEO ENCODER

In the AVSS task, video encoders face a critical trade-off: large-scale backbones achieve strong semantic alignment with lip movements but incur prohibitive computational costs, while lightweight encoders often fail to capture high-level semantic features and lead to suboptimal separation quality. To overcome this challenge, we design a dual-path autoencoder, DP-LipCoder (Figure 2), which extracts both reconstruction-related and semantic-related features from the video stream. The dual-path design is motivated by two complementary observations: (i) speaker images inherently contain auxiliary cues such as facial expressions and speaker identity, which are also crucial for speech separation; (ii) speech and lip motions exhibit strong temporal alignment.

Specifically, both the reconstruction path and the semantic path adopt the same encoder structure. However, The encoder parameters are not shared between the two paths. The encoder structure is adapted from video generation network MagVIT (Yu et al., 2023). The encoder consists of cascaded 3D residual blocks and spatial attention blocks, with alternating spatial downsampling. At the end of the semantic path, an additional single-step VQ module is introduced to extract discrete semantic features from semantic path encoder output $\mathbf{Z}_\mathrm{e}$. Subsequently, the outputs of the two encoders are denoted as $\mathbf{V}_\mathrm{r}$ and $\mathbf{V}_\mathrm{s}$, respectively. $\mathbf{V}_\mathrm{r}$ and $\mathbf{V}_\mathrm{s}$ are fused via summation and passed to the decoder, which mirrors the encoder structure. The decoder progressively reconstructs along the spatial dimension, yielding the final reconstructed video $\hat{\mathbf{V}} \in \mathbb{R}^{H \times W \times T_\mathrm{v}}$. Further architectural details are in Appendix A.1.

To guide the dual-path network toward simultaneously and effectively modeling $\mathbf{V}_\mathrm{r}$ and $\mathbf{V}_\mathrm{s}$, we introduce three losses during training (Figure 2): (i) Reconstruction loss ($\mathcal{L}_\mathrm{recon}$), which improves the fidelity of the reconstructed frames and encourages the reconstruction path to capture speaker-related cues. (ii) Distillation loss ($\mathcal{L}_\mathrm{distill}$), where AV-HuBERT is employed as a teacher model to guide the semantic path toward extracting audio-aligned semantic features. (iii) Commitment loss ($\mathcal{L}_\mathrm{commit}$) following the standard formulation in VQ (Esser et al., 2021), which penalizes discrepancies between encoder outputs and their nearest codebook entries. The details of different losses are provided in Appendix A.2. The overall training objective is

$$\mathcal{L} = \mathcal{L}_\mathrm{commit} + \mathcal{L}_\mathrm{distill} + \mathcal{L}_\mathrm{recon}. \tag{2}$$

Experimental settings and training results for the pre-training stage are in Appendices A.3 and A.4, respectively. In the AVSS task, we only infer the encoders and their VQ modules corresponding to the reconstruction and semantic paths, thereby obtaining the visual tokens $\mathbf{V}_r$ and $\mathbf{V}_s$.

### 3.3 AUDIO-VISUAL FUSION (AVF) MODULE

To achieve efficient AV fusion, the AVF module of Dolphin adopts two fusion mechanisms proposed in RTFSNet (Pegg et al., 2023): video-guided gated fusion $\mathcal{F}_1$ and attention-based fusion across multiple visual feature spaces $\mathcal{F}_2$. Furthermore, we extend these mechanisms from the time-frequency domain to time domain approach. Specifically, we perform upsampling solely along the temporal dimension of the visual features, thereby avoiding redundant expansion in the frequency dimension. Finally, the AVF module aggregates the features obtained from both fusion pathways through element-wise summation, generating the output representation $\mathbf{F} \in \mathbb{R}^{N_a \times T_a}$. A detailed description of the complete fusion process and implementation can be found in Appendix B.

### 3.4 SEPARATOR

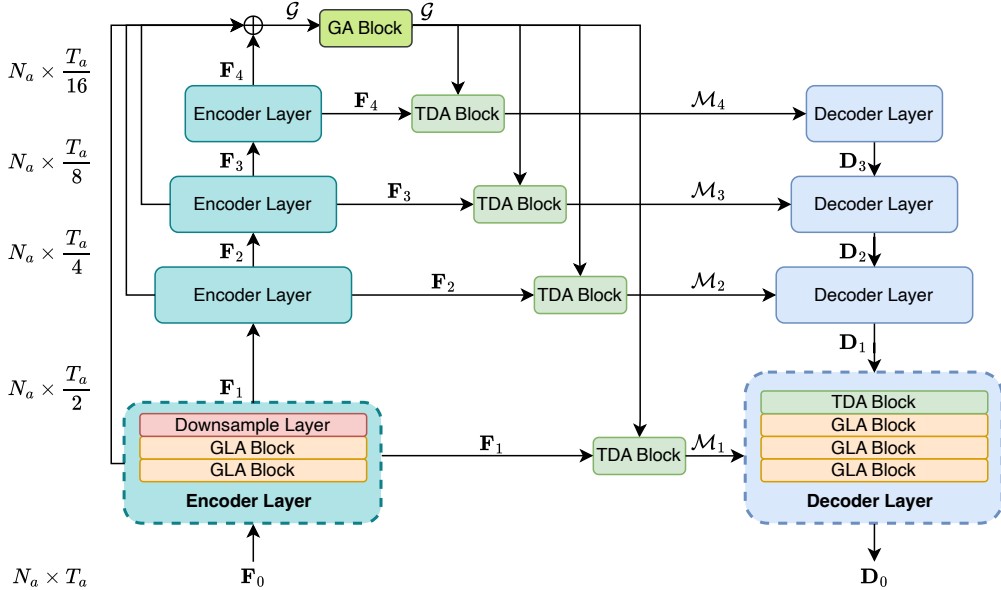

Figure 3: The architecture of separator. Here, we set the encoder and decoder layers $Q$ to 4.

Encoder-decoder architectures have demonstrated strong performance in speech separation tasks due to their ability to extract and integrate multi-scale features (Li et al., 2023; Xu et al., 2025). Motivated by this, we adopt an audio-only speech separation network TDANet (Li et al., 2023) as the backbone of our separator, as shown in Figure 3. It is worth noting that the original TDANet leverages multiple iterative separators to progressively refine the separation results, but this design considerably increases inference time. To improve efficiency, we retain only a single separator iteration while compensating for potential performance degradation by jointly modeling global and local attention in each layer (Shin et al., 2024). This design achieves high-quality separation with substantially reduced computational cost. Specifically, the encoder is composed of $Q$ stacked layers, where each layer consists of two global-local attention (GLA) blocks followed by a downsampling layer. At the top layer, we employ the GA block within a GLA block (see Figure 4(a)). Symmetrically, the decoder is built with $Q$ layers, each consisting of a top-down attention (TDA) block from TDANet that performs upsampling, followed by three stacked GLA blocks.

#### 3.4.1 GLA BLOCK

The GLA block (see Figure 4) is designed to enhance audio representations by combining global and local feature modeling, thereby reducing the dependence of the separator on multiple iterations. We describe the first GLA block in the first layer of the encoder, as all GLA blocks are identical.

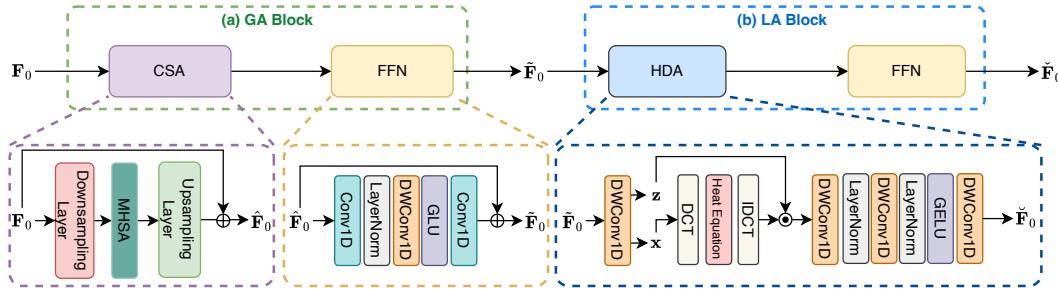

Figure 4: Detailed architectures of GLA blocks in the separator.

**GA Block.** We design a GA block consisting of a CSA layer followed by a FFN. Crucially, the multi-head self-attention (MHSA) mechanism within the CSA layer operates across the entire time dimension of the input sequence. This allows the model to effectively capture global contextual features and long-range dependencies. To mitigate the quadratic complexity of MHSA along the temporal dimension, the CSA layer first downsamples $\mathbf{F}_0$ to length $T_a/2^Q$, applies a multi-head self-attention function $f_{\mathrm{MHSA}}$ to capture global dependencies, and then upsamples the result back to the original length, producing $\hat{\mathbf{F}}_0$. This design reduces the computational complexity of MHSA to $1/2^{2Q}$ of the original. Subsequently, the FFN leverages stacked convolutional operations to further refine the $\hat{\mathbf{F}}_0$, as shown in Figure 4(a). In particular, depthwise convolution (DWConv1D) with kernel size 3 is employed. The final output is the globally enhanced representation $\tilde{\mathbf{F}}_0 \in \mathbb{R}^{N_a \times T_a}$.

**LA Block.** Although the GA block demonstrates strong capabilities in modeling long-range contextual features, it still exhibits certain limitations in capturing the local contextual structures of audio features. To enhance the model's ability to capture local structural patterns, we introduce an LA block, composed of one HDA layer and one FFN, following the GA block (see Figure 4(b)). The design motivation for HDA layer arises from the following observation: after the feature sequence is projected into a pseudo-frequency domain via the DCT transformation, local features can be decomposed into different frequency components (Denis et al., 2002; Imtiaz & Fattah, 2013). Therefore, integrating various frequency components allows for precise modeling of local features.

To efficiently integrate these frequency components, we employ an exponential decay function, derived from the heat diffusion equation (Cannon, 1984), to implement a learnable multi-scale filtering mechanism. Compared with large kernel convolutions, this approach no longer relies on a limited convolutional kernel receptive field and enables fine-grained modeling of local features (see Figure 5 in Appendix), without introducing a significant number of parameters. Subsequently, we reconstruct the filtered features and map them back to the time domain via an inverse transform. The key advantage of this design lies in the fact that the basic shape of the filter is constrained by physical priors, and the model only needs to learn a small number of scaling and gating parameters for each channel. As a result, the method not only substantially reduces the risk of overfitting but also achieves considerable computational efficiency.

Specifically, we first double the channel dimension of $\tilde{\mathbf{F}}_0$ through a convolutional projection and split the result into an initial condition $\mathbf{x} \in \mathbb{R}^{N_a \times T_a}$ and a gating signal $\mathbf{z} \in \mathbb{R}^{N_a \times T_a}$. We then apply the discrete cosine transform (DCT) (Strang, 1999) to map $\mathbf{x}$ into the frequency domain $\mathbf{A}(p) \in \mathbb{R}^{N_a}$, where $p \in \{0, 1, \ldots, T_a - 1\}$ denotes the frequency index:

$$\mathbf{A}(p) = \mathrm{DCT}[\mathbf{x}] = \sum_{t=0}^{T_a-1} \sqrt{\tfrac{2}{T_a}} \cos\left(\tfrac{p\pi(t+0.5)}{T_a}\right) \mathbf{x}_t. \qquad (3)$$

In the frequency domain, an adaptive smoothing process is applied to each channel following a heat diffusion mechanism. For frequency index $p$, the attenuation is given by:

$$\tilde{\mathbf{A}}(p) = \mathbf{A}(p) \cdot \exp\left(-\mathbf{k}_c\left(\tfrac{p\pi}{T_a}\right)^2\right), \tilde{\mathbf{A}}(p) \in \mathbb{R}^{N_a}, \qquad (4)$$

where $\mathbf{k}_c \in \mathbb{R}^{N_a}$ is a learnable, channel-adaptive diffusion coefficient controlling the degree of smoothing. The signal is then transformed back to the temporal domain using inverse DCT (IDCT).

Finally, the output of the HDA block $\breve{\mathbf{F}}_0 \in \mathbb{R}^{N_a \times T_a}$ is derived through a gating mechanism and a projection layer:

$$\breve{\mathbf{F}}_0 = \mathcal{P}(\hat{\mathbf{x}} \odot \mathrm{SiLU}(\mathbf{z})), \tag{5}$$

where $\mathcal{P}(\cdot)$ denotes the multi-layer depthwise convolutional network, and SiLU is the activation function (Elfwing et al., 2018). Finally, a FFN identical to that employed in the GA block is applied for feature transformation, yielding the output of the LA block $\breve{\mathbf{F}}_0 \in \mathbb{R}^{N_a \times T_a}$.

### 3.4.2 ENCODER OF SEPARATOR

The encoder in the separation network employs multiple GLA blocks to efficiently extract and aggregate multi-scale features across different temporal resolutions. The core idea is to progressively capture audio-visual representations at different temporal granularities through layer-wise downsampling. Specifically, after $q$ stages of downsampling, the output features can be denoted as $\{\mathbf{F}_q \in \mathbb{R}^{N_a \times \frac{T_a}{2^q}} \mid q \in [1, 2, \ldots, Q]\}$. These multi-scale features are further downsampled to the lowest resolution $T_a/2^Q$, followed by element-wise summation to obtain a global audio-visual representation $\mathcal{G} \in \mathbb{R}^{N_a \times \frac{T_a}{2^Q}}$. This global representation $\mathcal{G}$ is then fed into the top-level GA block, which enhances its representational power and yields refined global features $\bar{\mathcal{G}} \in \mathbb{R}^{N_a \times \frac{T_a}{2^Q}}$. Subsequently, both $\bar{\mathcal{G}}$ and the encoder outputs at different scales $\mathbf{F}_q$ are input into the TDA blocks (Li et al., 2023). In this process, the global representation serves as a guiding signal to modulate local features, producing new multi-scale features $\mathcal{M}_q \in \mathbb{R}^{N_a \times \frac{T_a}{2^q}}$.

### 3.4.3 DECODER OF SEPARATOR

The decoder reconstructs the audio features of the target speaker through top-down connections. Its core is composed of TDA and GLA blocks, whose implementations are consistent with those in the encoder layers. Unlike prior approaches (Li et al., 2024a;b; Zhao et al., 2025), our decoder directly outputs the target speaker features, without relying on multiplying mixture features by a mask (Wang et al., 2023). After completing feature extraction from the highest layer to the lowest, the decoder output is passed through an output layer consisting of a $1 \times 1$ convolution, a GLU activation, and another $1 \times 1$ convolution, yielding the final representation $\mathbf{E} \in \mathbb{R}^{N_a \times T_a}$.

## 4 EXPERIMENT CONFIGURATIONS

**Dataset.** Following previous works (Li et al., 2024a; Wu et al., 2019; Li et al., 2024b), we conducted a systematic evaluation of the proposed model, alongside several baselines, on three widely adopted AVSS benchmarks: LRS2 (Afouras et al., 2018a), LRS3 (Afouras et al., 2018c), and VoxCeleb2 (Chung et al., 2018). Unless otherwise specified, the training and evaluation uses 2-second speech segments (16 kHz sampling rate), with audio from two speakers mixed by default. Visual and audio are strictly synchronized, with a frame rate of 25 FPS, and the visual input consists of $88 \times 88$ grayscale lip images. More details on the datasets can be found in Appendix C.

**Model Configurations and Evaluation Metrics.** We used the Adam optimizer (Kingma & Ba, 2015) for training with an initial learning rate of $1 \times 10^{-3}$. When the validation loss plateaued for 15 epochs, the learning rate was halved; if it stalled for more than 30 epochs, early stopping was triggered. To prevent gradient explosion, we applied L2 gradient clipping with a threshold of 5, and we used the SI-SNR (Le Roux et al., 2019) as the optimization objective, with details provided in Appendix D. All training was conducted on 8 RTX 5090 GPUs with a batch size of 48. Detailed hyperparameter configurations are provided in Appendix E. For evaluation, we adopted SDRi (Vincent et al., 2006) and SI-SNRi as the main separation metrics, and additionally introduced PESQ (Union, 2007) to assess the speech quality. To measure efficiency, we report the model parameters, MACs, CPU/GPU inference latency, and memory usage, all tested on 1-second audio.

## 5 RESULTS

### 5.1 EFFECTIVENESS OF THE PRETRAINED VIDEO ENCODER

To verify the effectiveness of the DP-LipCoder, we conducted a separation comparison with pretrained video encoders on LRS2 dataset (Afouras et al., 2018a). Specifically, the baselines include:

(i) 3D ResNet-18 (Ma et al., 2021) pretrained on lip-reading task; (ii) Autoencoder (AE) (Martel et al., 2023) pretrained on single-frame reconstruction task; (iii) LipCoder, a variant of DP-LipCoder by removing the reconstruction path to examine the effectiveness of the semantic path; and (iv) DP-LipCoder, incorporating the reconstruction and semantic paths (see Figure 2). For a detailed introduction to AE and 3D ResNet-18, see Appendix F.

Table 1: Comparison of pretrained video encoder methods on the LRS2 dataset. **Best results** are in bold, with second-best results underlined. SI-SNRi and SDRi are measured in dB. Parameter count and computational cost (MACs) are calculated for the pretrained video encoder.

| Methods | SI-SNRi ↑ | SDRi ↑ | PESQ ↑ | Params (MB) ↓ | MACs (G/s) ↓ | Latency (ms) ↓ | |
| --- | --- | --- | --- | --- | --- | --- | --- |
| | | | | | | CPU | GPU |
| 3D ResNet-18 | **17.0** | **17.1** | **3.30** | 11.19 | 7.95 | 2908.26 | 33.10 |
| AE | 15.2 | 15.4 | 3.15 | **0.05** | **0.17** | **2082.04** | **22.92** |
| LipCoder | 16.3 | 16.4 | 3.24 | 0.65 | 5.33 | 2473.84 | 25.53 |
| DP-LipCoder | 16.8 | 16.9 | 3.29 | 0.78 | 2.38 | 2117.96 | 23.24 |

As shown in Table 1, the results confirm the effectiveness of introducing discrete visual features for AVSS. Both LipCoder and DP-LipCoder leverage the advantages of discrete encoding, achieving clear gains over the continuous autoencoder baseline, with SI-SNRi improvements of at least 1.0 dB. This demonstrates that quantizing visual signals into a discrete "visual vocabulary" encourages more compact and discriminative representations, thereby enhancing separation performance. In addition, the proposed method is highly efficient compared with existing SOTA systems. Taking 3D ResNet-18 as a reference, DP-LipCoder attains a comparable SI-SNRi (within 0.2 dB) while reducing parameters by 93% and MACs by 70%. These findings highlight the strong trade-off between performance and efficiency enabled by our discretization scheme, offering new insights for lightweight yet effective AV system design. Based on this balance, we adopted the DP-LipCoder for visual feature extraction. Further ablation studies confirm the critical role of the VQ module, which contributes approximately 0.5 dB to the SI-SNRi by enhancing semantic alignment. Detailed results and analysis are provided in Appendix G.

Table 2: Performance comparison of AVSS methods with pretrained video encoders replaced by DP-LipCoder on the LRS2 dataset. All results were obtained by retraining the AVSS model from scratch with the frozen DP-LipCoder. Relative changes post-replacement are shown in parentheses, with red indicating improvements and green indicating degradation. Parameter count and computational cost (MACs) represent the combined totals of the pretrained video encoder and AVSS model.

| Methods | SI-SNRi ↑ | SDRi ↑ | PESQ ↑ | Params (MB) ↓ | MACs (G/s) ↓ |
| --- | --- | --- | --- | --- | --- |
| CTCNet (Li et al., 2024a) | $14.0_{(-0.3)}$ | $14.4_{(-0.2)}$ | $3.07_{(-0.01)}$ | $7.65_{(-10.41)}$ | $78.18_{(-5.57)}$ |
| RTFSNet (Pegg et al., 2023) | $14.8_{(-0.1)}$ | $15.1_{(0.0)}$ | $3.07_{(0.0)}$ | $1.63_{(-10.41)}$ | $35.94_{(-5.57)}$ |
| IIANet (Li et al., 2024b) | $15.4_{(-0.6)}$ | $15.6_{(-0.6)}$ | $3.18_{(+0.05)}$ | $4.60_{(-10.41)}$ | $20.94_{(-5.57)}$ |
| AV-Mossformer2 (Zhao et al., 2025) | $15.0_{(-0.1)}$ | $15.3_{(-0.2)}$ | $3.16_{(0.0)}$ | $58.50_{(-10.02)}$ | $117.87_{(-6.59)}$ |

**Generalization to Existing Separation Architectures:** To evaluate the generalization ability of DP-LipCoder, we replaced the original video encoders in several AVSS models with our DP-LipCoder, while keeping the rest of the models unchanged. As shown in Table 2, although DP-LipCoder results in a slight performance drop, the substantial efficiency gain makes it acceptable for practical deployment. In particular, under resource-constrained scenarios, DP-LipCoder provides an efficient visual encoding solution.

## 5.2 COMPARISON WITH STATE-OF-THE-ART METHODS

We conducted extensive experiments to quantitatively compare the proposed Dolphin with existing methods on three datasets. The results are shown in Table 3. For fairness, all baseline scores were either extracted from the original papers or reproduced with officially released implementations.

**Separation Performance.** We conducted comprehensive evaluations of Dolphin's separation capabilities across three benchmark datasets and compared against SOTA methods. Table 3 demonstrates that Dolphin consistently surpasses all competing approaches across all evaluation metrics. We further examined multi-speaker scenarios using the LRS2-3Mix and LRS2-4Mix datasets (Li et al.,

Table 3: Performance comparison of various AVSS methods. Results for AV-Mossformer2 and Swift-Net were reproduced using the official code, while results for other methods are reported directly from their original papers.

| Methods | LRS2 | | | LRS3 | | | VoxCeleb2 | | |
|---|---|---|---|---|---|---|---|---|---|
| | SI-SNRi ↑ | SDRi ↑ | PESQ ↑ | SI-SNRi ↑ | SDRi ↑ | PESQ ↑ | SI-SNRi ↑ | SDRi ↑ | PESQ ↑ |
| AV-ConvTasNet (Wu et al., 2019) | 12.5 | 12.8 | 2.69 | 11.2 | 11.7 | 2.58 | 9.2 | 9.8 | 2.17 |
| Visualvoice (Gao & Grauman, 2021) | 11.5 | 11.8 | 2.78 | 9.9 | 10.3 | 2.13 | 9.3 | 10.2 | 2.45 |
| AVLiT-8 (Martel et al., 2023) | 12.8 | 13.1 | 2.56 | 13.5 | 13.6 | 2.78 | 9.4 | 9.9 | 2.23 |
| CTCNet (Li et al., 2024a) | 14.3 | 14.6 | 3.08 | 17.4 | 17.5 | 3.24 | 11.9 | 13.1 | 3.00 |
| RTFS-Net (Pegg et al., 2023) | 14.9 | 15.1 | 3.07 | 17.5 | 17.6 | 3.25 | 12.4 | 13.6 | 3.00 |
| IIANet (Li et al., 2024b) | 16.0 | 16.2 | 3.23 | 18.3 | 18.5 | 3.28 | 13.6 | 14.3 | 3.12 |
| AV-Mossformer2 (Zhao et al., 2025) | 15.1 | 15.5 | 3.16 | 17.7 | 18.1 | 3.28 | 14.0 | 14.6 | 3.13 |
| Swift-Net (Sang et al., 2025) | 13.9 | 14.2 | 3.07 | 15.8 | 16.4 | 3.11 | 12.8 | 13.5 | 2.99 |
| Dolphin (*ours*) | **16.8** | **16.9** | **3.29** | **18.8** | **18.9** | **3.36** | **14.6** | **15.1** | **3.17** |

2024b), with detailed results presented in Appendix H. Dolphin demonstrates consistent improvements over existing methods, confirming the generalization capability of our approach. To further validate robustness, we provide evaluations under diverse noise conditions in Appendix I, alongside a subjective MOS evaluation on real overlapping speech in Appendix J, which corroborates the superiority of Dolphin in in-the-wild scenarios. Audio samples in the demo page [1] reveal that Dolphin produces noticeably superior clarity compared to baseline methods, particularly under substantial background noise. Additionally, Appendix K provides spectrogram visualizations illustrating the separation performance of Dolphin.

Table 4: Efficiency comparison of AVSS methods. "w/o" and "w/" denote without and with pretrained video encoder, respectively. "RAM" indicates system memory usage during CPU inference; "GPU Inf." indicates GPU memory usage during inference.

| Methods | Parameters (M)↓ | | MACs (G)↓ | | Latency (ms)↓ | | Memory (MB)↓ | |
|---|---|---|---|---|---|---|---|---|
| | w/o | w/ | w/o | w/ | CPU | GPU | RAM | GPU Inf. |
| AV-ConvTasNet (Wu et al., 2019) | 13.70 | 25.01 | **6.94** | 14.90 | 342.15 | 10.54 | 95.47 | 129.51 |
| Visualvoice (Gao & Grauman, 2021) | 68.61 | 77.75 | 16.82 | 18.20 | **280.29** | **7.19** | 296.84 | 338.96 |
| AVLiT-8 (Martel et al., 2023) | 5.54 | **5.57** | 17.52 | 17.53 | 547.77 | 23.31 | **21.23** | **52.09** |
| CTCNet (Li et al., 2024a) | 6.87 | 18.06 | 75.80 | 83.75 | 1433.68 | 37.52 | 68.95 | 138.42 |
| RTFS-Net (Pegg et al., 2023) | **0.85** | 12.03 | 33.56 | 41.51 | 4021.93 | 56.59 | 48.40 | 249.49 |
| IIANet (Li et al., 2024b) | 3.82 | 15.01 | 18.56 | 26.51 | 3213.82 | 142.30 | 57.31 | 148.14 |
| AV-Mossformer2 (Zhao et al., 2025) | 55.87 | 68.52 | 111.79 | 124.46 | 3083.86 | 62.30 | 262.45 | 398.76 |
| Swift-Net (Sang et al., 2025) | 1.47 | 12.66 | 18.24 | 26.18 | 2589.76 | 39.45 | 48.33 | 244.53 |
| Dolphin (*ours*) | 6.22 | 7.00 | 8.51 | **10.89** | 2117.96 | 33.24 | 26.84 | 251.12 |

**Separation Efficiency.** Computational efficiency is crucial for real-world deployment of AVSS models. Unlike prior studies that overlooked computational overhead of pretrained video encoders, we comprehensively evaluated both audio and visual components. Table 4 demonstrates that Dolphin achieves superior efficiency while maintaining strong separation quality. Compared to SOTA IIANet, Dolphin required only 46% of the MACs (excluding pretrained video encoder), and its CPU inference time was reduced to 66%. Against lightweight alternatives Swift-Net and RTFS-Net, Dolphin reduced inference time by 18% and 47%, respectively. When incorporating pretrained video encoder, Dolphin achieved the lowest MACs among all methods.

## 5.3 ABLATION STUDY

To evaluate the effectiveness of each key component in Dolphin, we conducted ablation studies on the LRS2 dataset, while keeping all other training and evaluation settings identical to the full model.

**Contribution of Components in GLA Block.** The proposed GLA block enhances audio representations by combining global context and local details. To assess each component's role, we evaluated four variants: full GLA (Dolphin), without LA, without GA, and without both (the omitted LA/GA modules are substituted by a single depthwise convolution). Results in Table 5 show that removing either part degrades performance, and removing both results in the worst outcome. This confirms that global and local modeling are complementary and jointly crucial for effective speech separation.

---

[1] https://cslikai.cn/Dolphin

Table 5: Ablation study on the contribution of GA and LA blocks in the proposed GLA block.

| GA | LA | SI-SNRi↑ | SDRi↑ | PESQ↑ | Params (MB)↓ | MACs (G/s)↓ |
|---|---|---|---|---|---|---|
| × | × | 10.4 | 10.5 | 2.43 | 2.04 | 3.29 |
| ✓ | × | 15.9 | 16.1 | 3.20 | 5.23 | 7.59 |
| × | ✓ | 15.6 | 15.8 | 3.18 | 3.81 | 6.59 |
| ✓ | ✓ | **16.8** | **16.9** | **3.29** | **7.00** | **10.89** |

Table 6: Performance comparison of model with and without HDA layer.

| Methods | SI-SNRi↑ | SDRi↑ | PESQ↑ | Params (MB)↓ | MACs (G/s)↓ |
|---|---|---|---|---|---|
| HDA layer | **16.9** | **16.8** | **3.29** | **7.00** | **10.89** |
| Conv1D | 16.5 | 16.4 | 3.25 | 7.57 | 11.19 |

**HDA Layer.** To verify the effectiveness of the HDA layer, we compare it with an alternative design where the local feature modeling component is replaced by a one-dimensional large-kernel convolution. Specifically, the latter models local features purely through convolution, whereas the former explicitly incorporates physical priors via a heat diffusion mechanism. As shown in Table 6, the HDA layer achieves consistent improvements across all three metrics, while also reducing the number of learnable parameters during computation.

To further clarify the effectiveness of our design choices, we conducted a series of ablation studies. First, we compared our single-iteration design against iterative refinement (Appendix L). The results show that our single-iteration Dolphin delivers superior separation accuracy with similar computational cost, while avoiding the overhead of multi-iteration methods, underscoring the efficiency of the GLA module. Building on this core design, we examined additional architectural variants. We determined that a relatively deeper decoder leads to improved reconstruction quality (Appendix M), and that direct feature regression mitigates distortions from conventional masking (Appendix N). Additionally, we confirmed that early-stage audio-visual fusion in the separator's encoder is optimal, boosting SI-SNRi by $0.6\,\text{dB}$ compared to later fusion (Appendix O).

## 6 CONCLUSIONS

In this study, we propose an efficient AVSS model, Dolphin, designed to address the long-standing trade-off between computational efficiency and separation performance in existing approaches. To alleviate the parameter redundancy introduced by conventional visual backbones, we design the DP-LipCoder, which maps lip videos into discrete semantic units that are closely aligned with audio. For the audio network, we design a lightweight encoder–decoder separator and incorporate GLA blocks to sequentially capture both global and local features, enabling the separator to achieve high-quality audio separation in a single iteration. Extensive experiments on three benchmark datasets demonstrate that Dolphin achieves a favorable balance between performance and efficiency, offering a practical and deployable solution for AVSS systems in real-world scenarios.

Limitations and Future Work. While Dolphin balances efficiency and performance, several avenues remain for exploration. One limitation lies in the reliance on relatively clean and synchronized lip videos; future work should enhance robustness against large head pose variations, occlusions, and extreme lighting conditions. Additionally, although our model significantly reduces computational costs compared to prior AVSS systems, deployment on extremely resource-constrained edge devices remains a challenge. We plan to investigate techniques like quantization or pruning to bridge this gap. Finally, while mapping lip motion to discrete tokens improves efficiency, it may discard fine-grained articulatory cues. Exploring hierarchical codebooks or mixed discrete–continuous representations offers a promising direction to recapture this subtle visual information.

ACKNOWLEDGMENTS

This work was supported in part by the National Key Research and Development Program of China (No. 2021ZD0200301), the National Natural Science Foundation of China (Nos. U2341228 and

62576187), and the Fundamental and Interdisciplinary Disciplines Breakthrough Plan of the Ministry of Education of China (No. JYB2025XDXM504).

REPRODUCIBILITY STATEMENT

The implementation of Dolphin is developed in Python 3.11, relying on standard deep learning libraries, in particular PyTorch and PyTorch Lightning. To ensure full reproducibility, we will release the code of Dolphin under the Apache-2.0 license on GitHub once this work is accepted for publication. The repository will include all necessary files for reproducing the experiments (see Table 3), such as the conda environment specification, complete configuration files, pretrained weights for the video backbone, and the source code of Dolphin. Datasets need to be obtained independently following the cited references (Appendix C), as they are proprietary to their respective publishers, but we will provide preprocessing scripts in the repository. A PyTorch implementation of VQ, maintained by its original authors, is publicly available on PyPi[2]. All experiments and training procedures were conducted on a server equipped with eight NVIDIA 5090 GPUs; detailed specifications are reported in Section 4 and Appendix A.3. For researchers aiming to reproduce our results, all hyperparameters are listed in Appendix E. Evaluation metrics and loss functions are described with formal definitions in Appendix D, and will also be provided in the released codebase [3].

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

APPENDIX

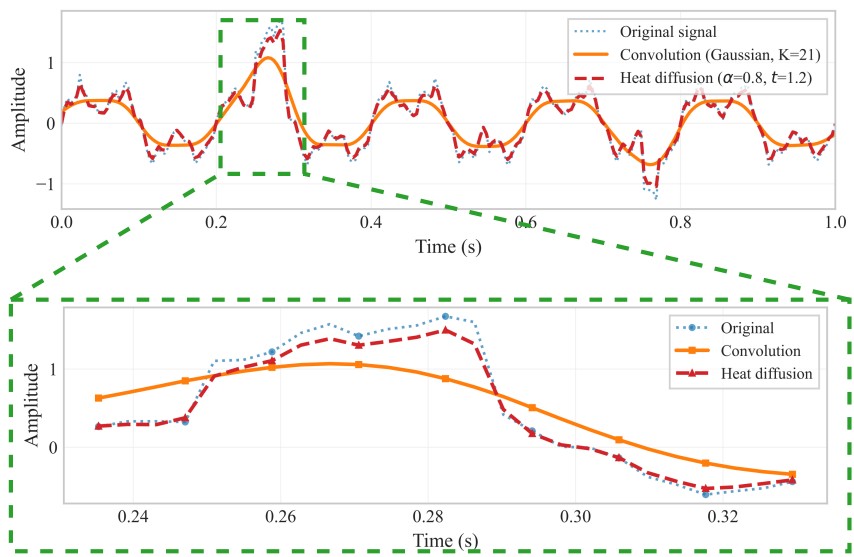

Figure 5: Comparison of heat diffusion filtering (red dashed, $\alpha = 0.8, t = 1.2$) and Gaussian convolution (orange solid, K = 21) applied to a test signal containing multiple frequency components and local impulses (blue dotted). The inset shows a magnified view of the impulse response region (green dashed rectangle), revealing superior edge preservation by the heat diffusion method while maintaining effective noise suppression.

## A  DETAILS OF DP-LIPCODER NETWORK

### A.1  VIDEO AUTOENCODER ARCHITECTURE

In this section, we provide a detailed description of the implementation of the DP-LipCoder. The decoder is structurally symmetric to the encoder, where each downsampling operation is replaced with the corresponding upsampling operation to reconstruct the features.

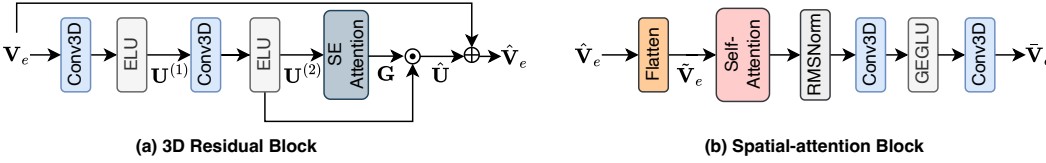

Figure 6: Detailed architecture of the 3D residual block and the spatial-attention block.

As illustrated in Figure 2, we design the encoder to project the input video clip $\mathbf{V}$ into a compact spatiotemporal representation, while alternately performing feature transformation and spatial compression across multiple scales. First, a 3D convolutional layer is applied to expand the input channels, producing $\mathbf{V}_e \in \mathbb{R}^{N_v \times H \times W \times T_v}$. Unless otherwise stated, the formulas are illustrated using features at the highest spatial resolution; the same computation applies to other resolution levels, which differ only in spatial size. At each resolution level, we employ multiple 3D residual blocks $f_r(\cdot)$ to extract spatiotemporal features, as shown in Figure 6(a). Each residual unit consists of "two 3D convolutional layers + SE attention + residual connection", and its computation can be formally expressed as follows:

(1) Two consecutive 3D convolutions with point-wise nonlinearity are applied as follows:

$$\mathbf{U}^{(1)} = \text{ELU}(\text{Conv3D}_{3\times3\times3}(\mathbf{V}_e)) \in \mathbb{R}^{N_v \times H \times W \times T_v}, \tag{6}$$

$$\mathbf{U}^{(2)} = \text{ELU}\Big(\text{Conv3D}_{1\times1\times1}(\mathbf{U}^{(1)})\Big) \in \mathbb{R}^{N_v \times H \times W \times T_v}. \tag{7}$$

(2) SE attention (Hu et al., 2018) (frame-wise spatial aggregation) operates as follows. First, a per-frame $1 \times 1$ 2D convolution is applied to produce the spatial attention logits $\mathbf{l}_t$, which are normalized into weights $\boldsymbol{\alpha}_t$ by a spatial softmax. Each channel is then aggregated with these weights to obtain a global context vector $\mathbf{s}_t$. Subsequently, a two-layer channel-wise feed-forward network generates the gating coefficients $\mathbf{g}_t$, which are broadcast across spatial dimensions to form the channel gating tensor $\mathbf{G} \in \mathbb{R}^{N_v \times H \times W \times T}$. The entire procedure can be formalized as:

$$\mathbf{l}_t = \text{Conv2D}_{1\times1}\Big(\mathbf{U}^{(2)}_{::t}\Big) \in \mathbb{R}^{1 \times H \times W}, \quad t = 1, \ldots, T_v, \tag{8}$$

$$\boldsymbol{\alpha}_t = \text{softmax}\big(\text{vec}(\mathbf{l}_t)\big) \in \mathbb{R}^{H \times W}, \tag{9}$$

$$\mathbf{s}_t(c) = \sum_{p \in \Omega} \boldsymbol{\alpha}_t(p)\, \mathbf{U}^{(2)}_c(p, t), \quad c = 1, \ldots, N_v, \tag{10}$$

$$\mathbf{g}_t = \sigma\big(\mathbf{W}_2\, \varrho(\mathbf{W}_1 \mathbf{s}_t)\big) \in \mathbb{R}^{N_v}, \tag{11}$$

$$\mathbf{G}_c(p, t) = \mathbf{g}_t(c), \quad \forall p \in \Omega, \tag{12}$$

where $\Omega$ denotes the set of spatial positions, $\varrho(\cdot)$ is the LeakyReLU and the projection matrices are $\mathbf{W}_1 \in \mathbb{R}^{d_h \times N_v}$, $\mathbf{W}_2 \in \mathbb{R}^{N_v \times d_h}$, with hidden dimension $d_h = \lfloor N_v/2 \rfloor$.

(3) We apply channel gating to modulate the intermediate features:

$$\widehat{\mathbf{U}} = \mathbf{G} \odot \mathbf{U}^{(2)} \in \mathbb{R}^{N_v \times H \times W \times T_v}, \tag{13}$$

$$f_r(\mathbf{V}_e) = \widehat{\mathbf{U}} + \mathbf{V}_e \in \mathbb{R}^{N_v \times H \times W \times T_v}. \tag{14}$$

In this way, a 3D residual block integrates local spatiotemporal modeling with global context–aware channel recalibration. The residual connection further stabilizes training in deep networks and mitigates degradation.

This residual unit is placed within a multi-scale architecture, cooperating with subsequent spatial compression/reconstruction modules.

In the spatial-attention block (see Figure 6(b)), we first reshape the visual feature map from the 3D spatiotemporal representation $\hat{\mathbf{V}}_e \in \mathbb{R}^{N_v \times H \times W \times T_v}$ into a 2D sequence representation $\tilde{\mathbf{V}}_e \in \mathbb{R}^{N_v \times (H \times W) \times T_v}$. Subsequently, we apply the standard self-attention mechanism (Vaswani et al., 2017) on this sequence to capture long-range dependencies between different spatial positions within the same time frame, thereby enhancing the model's ability to represent spatial structural information. After the spatial-attention block, we further introduce a feed-forward network to enhance feature representation capability. This network mainly consists of a normalization layer and a pointwise convolution structure: first, a normalization layer (Zhang & Sennrich, 2019) is applied to the input features to mitigate distribution differences and stabilize training; then, a pointwise 3D convolution layer maps the channel number to an expanded dimension, and a gating unit (GEGLU) (Shazeer, 2020) is used to introduce higher-order nonlinear representations; finally, another point-wise 3D convolution layer maps the channel number back to the original dimension, resulting in the block output $\bar{\mathbf{V}}_e \in \mathbb{R}^{N_v \times H \times W \times T_v}$.

On the encoder side, we perform progressive compression by spatial downsampling to enhance semantic abstraction. Specifically, at each time step the feature tensor is regarded as a 2D image and compressed along the spatial dimensions using a $3 \times 3$ 2D convolution with stride 2, thereby halving the spatial resolution while preserving the temporal dimension $T$. On the decoder side, a mirror-symmetric structure recovers spatial resolution, where each downsampling step is replaced by sub-pixel upsampling (Shi et al., 2016). This design jointly ensures computational efficiency, spatial detail fidelity, and global consistency during reconstruction.

## A.2 Objective Function of the Pre-training Stage

During pre-training stage, to ensure reconstructability while aligning semantic information, we unify the reconstruction, quantization-commitment, and semantic distillation objectives into a single end-

to-end multi-task loss within the VQ-VAE framework. Let $\mathbf{V}$ be the input video. The encoder processes $\mathbf{V}$ to output continuous latent vectors $\mathbf{Z}_{\mathrm{e}}$, which are then mapped by a vector quantizer to a discrete latent representation $\mathbf{V}_{\mathrm{s}}$. Finally, a decoder uses $\mathbf{V}_{\mathrm{s}}$ and $\mathbf{V}_{\mathrm{r}}$ to produce the reconstructed video $\hat{\mathbf{V}}$.

Following standard VQ-VAE practice, we adopt symmetric codebook-update and commitment terms, where $\mathrm{sg}[\cdot]$ denotes the stop-gradient operator and $\beta$ is the commitment cost (set to 1.0):

$$\mathcal{L}_{\mathrm{commit}} = \|\mathrm{sg}[\mathbf{Z}_{\mathrm{e}}] - \mathbf{V}_{\mathrm{s}}\|_2^2 + \beta\|\mathbf{Z}_{\mathrm{e}} - \mathrm{sg}[\mathbf{V}_{\mathrm{s}}]\|_2^2, \tag{15}$$

where the first term updates the codebook embeddings toward the encoder outputs, and the second encourages the encoder outputs to remain close to the selected codebook entries.

To inject high-level AV semantics from a teacher model into the discrete representation, we adopt a pretrained AV-HuBERT as the teacher $\mathcal{T}$ and introduce a distillation head $\mathcal{D}$ on the post-quantization branch to predict semantic features aligned with the teacher's outputs, yielding the distillation loss:

$$\mathcal{L}_{\mathrm{distill}} = \|\mathcal{D}(\mathbf{Z}_{\mathrm{e}}) - \mathcal{T}(\mathbf{V})\|_2^2. \tag{16}$$

To stabilize training and directly optimize the fidelity of reconstructed lip images, we employ an $L_2$ reconstruction objective:

$$\mathcal{L}_{\mathrm{recon}} = \|\hat{\mathbf{V}} - \mathbf{V}\|_2^2. \tag{17}$$

The overall training objective is

$$\mathcal{L} = \mathcal{L}_{\mathrm{commit}} + \lambda_{\mathrm{distill}}\,\mathcal{L}_{\mathrm{distill}} + \lambda_{\mathrm{recon}}\,\mathcal{L}_{\mathrm{recon}}, \tag{18}$$

where $\lambda_{\mathrm{distill}}$ controls the relative weight of the distillation signal (set to 1.0 in our experiments), and $\lambda_{\mathrm{recon}}$ controls the relative weight of the reconstruction signal (also set to 1.0).

## A.3    EXPERIMENTAL SETUPS

### A.3.1    DATASETS

We constructed a unified dataset by integrating three publicly available audio-visual corpora: LRS2 (Afouras et al., 2018a), LRS3 (Afouras et al., 2018c), and VoxCeleb2 (Chung et al., 2018). For the raw video streams, we first perform face detection to locate and crop the lip region (Zhang et al., 2016), followed by an offline pre-extraction of lip motion sequences. The processed sequences are then segmented into fixed-duration clips. Each clip spans $2.0\,\mathrm{s}$ at a frame rate of $25\,\mathrm{Hz}$, yielding approximately $N = T \times f = 50$ frames. To ensure consistency with prior work (Li et al., 2024a;b), dataset partitioning strictly follows the official configurations of each corpus. Specifically, we merge the original training, validation, and testing subsets into corresponding unified splits, thereby preserving the original division of data.

### A.3.2    MODEL CONFIGURATION

The visual codec adopts a lightweight hierarchical architecture composed of residual and attention-based modules. The pipeline comprises stacked 3D residual blocks, spatial-attention blocks, and up/down-sampling layers, connected in the order shown in Figure 2. The input consists of single-channel grayscale, lip-cropped video frames with spatial resolution $88 \times 88$ and temporal resolution $25\,\mathrm{Hz}$. The channel width starts at $4$ and is progressively expanded to $32$ across stages. All convolutions are 3D: the input projection uses a $7 \times 7 \times 7$ kernel, the output reconstruction uses $3 \times 3 \times 3$, and the residual convolutions in intermediate layers use kernel size 3. All convolutional layers use constant padding. The attention module uses 8 heads with per-head dimension 32, and FlashAttention (Dao, 2024) can optionally be enabled to further improve the compute and memory efficiency of spatiotemporal modeling.

The discrete codebook is initialized in a data-driven manner via $k$-means. For each quantizer, $k$-means is run independently 10 times on the encoder features, and the resulting cluster centers are used as the initial entries of that quantizer's codebook. The default codebook size is 256 with embedding dimension 64. During training, stochastic code sampling (temperature $0.1$) is enabled to enhance exploration of the codebook space and mitigate the risk of mode collapse due to premature convergence.

### A.3.3 TRAINING DETAILS

We optimized Eq. (18) using the Adam optimizer with a learning rate of $1 \times 10^{-3}$. Training proceeded for up to 500 epochs with early stopping: it was terminated if the validation reconstruction loss does not improve for 20 consecutive epochs. Unless otherwise noted, all experiments were conducted on 4 NVIDIA RTX 3090 GPUs using distributed data parallel (DDP). The LipCoder and DP-LipCoder network were implemented in PyTorch, with a global batch size of 32 during training.

### A.4 RESULTS

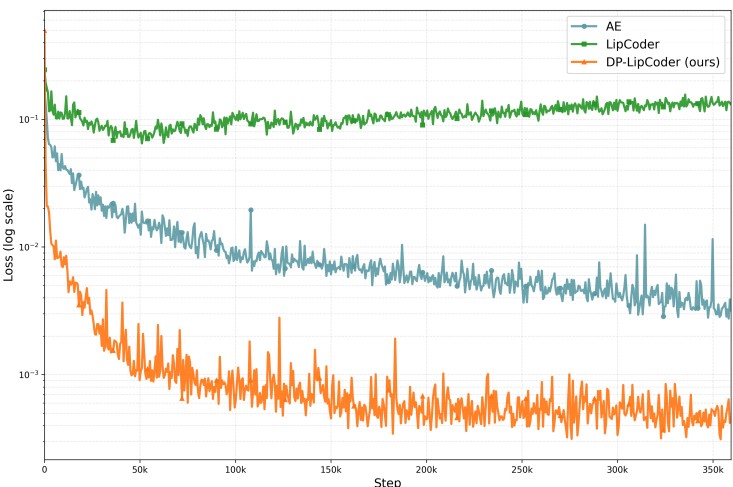

Figure 7: Reconstruction loss over training steps (log scale; lower is better).

To evaluate the reconstruction performance of our method, we trained an autoencoder (AE), a Lip-Coder, and DP-LipCoder on the same dataset. As shown in Figure 7, the three methods exhibit markedly different convergence behaviors and error floors during training. The LipCoder model without audio alignment remains in a high-loss regime throughout, reflecting codebook competition and drift when cross-modal semantic anchoring is absent; the decoder must compensate for semantic gaps with greater freedom, hindering error reduction. The AE trained solely with reconstruction loss shows a higher floor and sporadic spikes, indicating that its continuous latents primarily capture low-level appearance statistics rather than generalizable audio–visual structure. In contrast, DP-LipCoder unifies reconstruction, quantization–commitment, and semantic distillation into an end-to-end multi-task objective during pretraining, converging with lower variance to the smallest error—ultimately reducing error by about one order of magnitude relative to AE and by more than two orders relative to the unaligned VQ. These results indicate that explicitly coupling semantic alignment with reconstruction in a discrete bottleneck improves convergence speed, stability, and the achievable reconstruction quality, yielding more semantically consistent, reconstructible representations.

We further conducted a qualitative visualization analysis to examine the differences in reconstruction quality across methods. As shown in Figure 8, the misaligned LipCoder model exhibits evident structural omissions and blurry textures in its visual results. In contrast, the AE trained solely with reconstruction loss produces more coherent overall shapes than LipCoder. However, it still struggles to preserve fine-grained details and geometric consistency, often resulting in blurred contours or fragmented structures. Our proposed DP-LipCoder model demonstrates clear advantages both globally and locally: it preserves spatial structures more faithfully and generates textures with higher sharpness and realism. Beyond its superiority in quantitative metrics compared to AE and Lip-Coder, DP-LipCoder also achieves qualitatively better semantic consistency and structural fidelity, thereby providing comprehensive evidence of the effectiveness of the proposed approach. Additionally, since LipCoder explicitly encodes lip-motion videos into discrete representations, which better reflect the strong correlation between lip shapes and phonemes, it achieves effective separa-

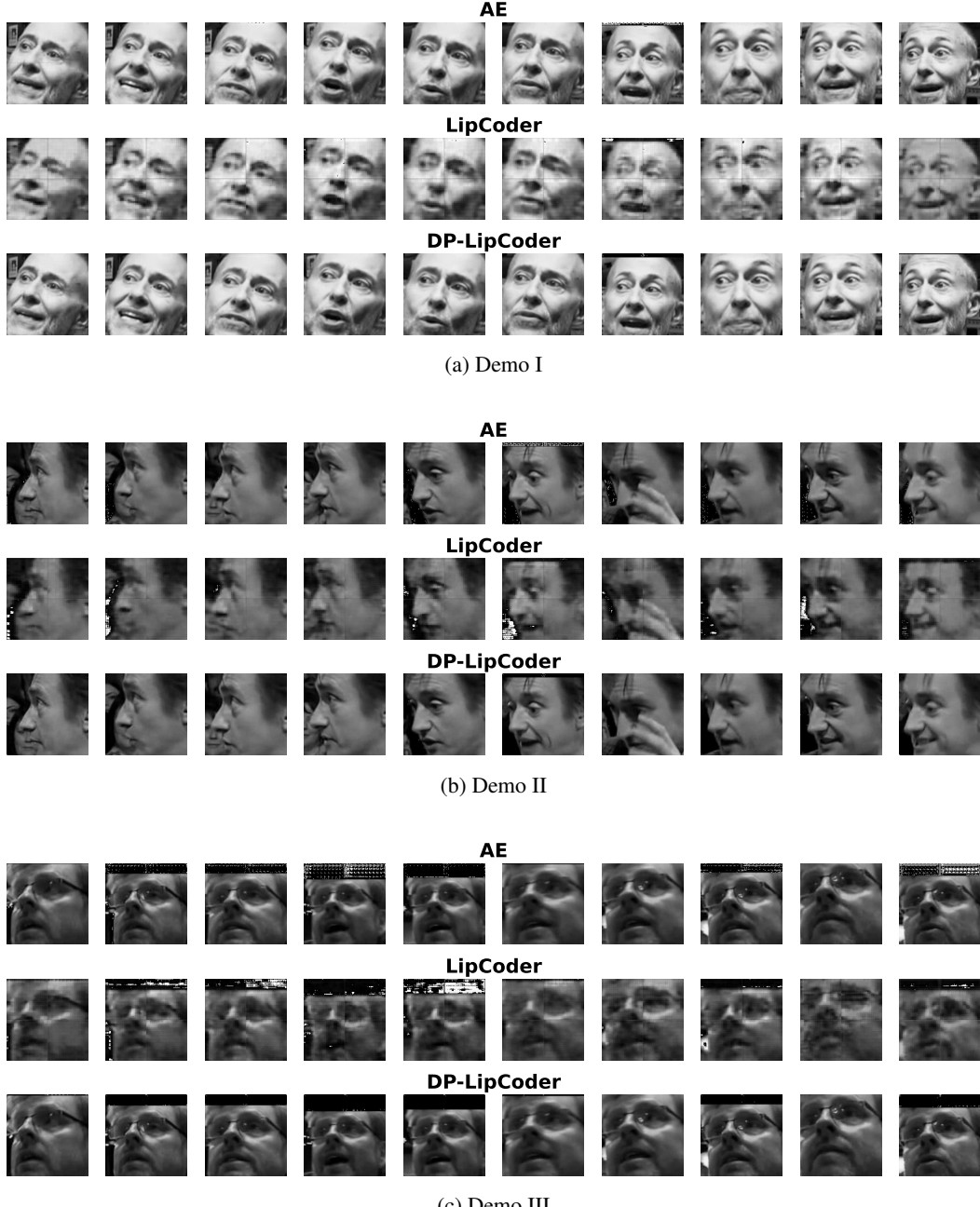

Figure 8: Visualization of reconstruction results obtained by different methods (AE, LipCoder, and the proposed DP-LipCoder) on three examples.

tion quality when serving as the pretrained video encoder even though its reconstruction quality is suboptimal.

# B    DETAILS OF AVF MODULE

Given the visual reconstruction tokens $\mathbf{V}_r$, semantic tokens $\mathbf{V}_s$, and audio features $\mathbf{X}$, we first integrate the two types of visual tokens with a U-Net-based visual feature fusion network (Pegg et al., 2023; Li et al., 2023), producing an aggregated visual representation $\tilde{\mathbf{V}} \in \mathbb{R}^{N_a \times T_v}$.

For the video-guided gated fusion, the semantic representation $\mathbf{V}_s$ is processed by a $1 \times 1$ depthwise convolution $W_1(\cdot)$, followed by temporal upsampling $\phi$, to generate the gating features. These are combined with audio features $\mathbf{X}_g$, obtained via another $1 \times 1$ depthwise convolution $W_2(\cdot)$, through an element-wise product $\odot$. This yields the video-guided gated fusion feature, expressed as

$$\mathcal{F}_1 = \phi(W_1(\tilde{\mathbf{V}})) \odot W_2(\mathbf{X}). \tag{19}$$

For multi-visual-space attention fusion, we decompose the visual representation into multiple sub-features, each guiding the fusion with audio features along different dimensions, thereby enhancing the fusion capability. Specifically, we expand the channel dimension of $\tilde{\mathbf{V}}$ with $1 \times 1$ depthwise convolution $W_3(\cdot)$ to obtain $\hat{\mathbf{V}} \in \mathbb{R}^{(N_a K) \times T_v}$. This is partitioned into $K$ sub-visual features $\tilde{\mathbf{v}}_k \in \mathbb{R}^{N_a \times T_v}$. The mean of these sub-features is passed through a softmax operation, followed by temporal upsampling, resulting in the multi-space attention representation $\check{\mathbf{V}} \in \mathbb{R}^{N_a \times T_a}$. Finally, $\check{\mathbf{V}}$ is fused with audio features processed by $W_4(\cdot)$, yielding

$$\begin{aligned}
\hat{\mathbf{V}} &= W_3(\tilde{\mathbf{V}}) = \left[\tilde{\mathbf{v}}_1; \tilde{\mathbf{v}}_2; \ldots; \tilde{\mathbf{v}}_K\right], \quad \tilde{\mathbf{v}}_k \in \mathbb{R}^{N_a \times T_v}, \\
\check{\mathbf{V}} &= \phi\left(\mathrm{Softmax}\left(\frac{1}{K}\sum_{k=1}^{K}\tilde{\mathbf{v}}_k\right)\right), \quad \check{\mathbf{V}} \in \mathbb{R}^{N_a \times T_a}, \\
\mathcal{F}_2 &= \check{\mathbf{V}} \odot W_4(\mathbf{X}), \qquad\qquad\quad \mathcal{F}_2 \in \mathbb{R}^{N_a \times T_a}.
\end{aligned} \tag{20}$$

Finally, the AVF module sums the two fused features, producing the output representation $\mathbf{F}$.

## C  DATASET DETAILS

To comprehensively evaluate the performance of our proposed method, we conducted experiments on three widely adopted benchmark datasets for the AVSS task: LRS2 (Afouras et al., 2018a), LRS3 (Afouras et al., 2018c), and VoxCeleb2 (Chung et al., 2018). These datasets naturally organize data from different speakers into separate directory structures, which facilitates the construction of speaker-independent evaluation paradigms.

To ensure fairness and reproducibility, we strictly followed the dataset partitioning protocols adopted by prior SOTA methods (Pegg et al., 2023; Li et al., 2024a;b). The detailed characteristics of each dataset are summarized as follows:

- **LRS2:** This dataset is collected from BBC television broadcasts and is characterized by numerous outdoor speaker segments. Visually, illumination conditions vary significantly, while acoustically, the audio often contains authentic background noise, thereby exhibiting strong real-world properties. Compared with earlier lip-reading datasets (Yang et al., 2019; Chung & Zisserman, 2016), LRS2 presents an unrestricted vocabulary and complete sentences, along with higher diversity in speaker identities and recording environments. The mixed version of LRS2 comprises approximately 11 hours of training data, 3 hours of validation data, and 1.5 hours of test data.

- **LRS3:** Constructed from over 400 hours of TED and TEDx talks available on YouTube, this dataset benefits from the professional nature of the recordings (e.g., speakers typically use microphones). As a result, the audio generally has a higher signal-to-noise ratio and a cleaner environment, complementing LRS2. The mixed subset used in our experiments contains 28 hours of training data, 3 hours of validation data, and 1.5 hours of test data.

- **VoxCeleb2:** Derived from large-scale "in-the-wild" YouTube videos, VoxCeleb2 presents highly challenging acoustic conditions, including non-stationary noise such as laughter, background conversations, and variable reverberations, which create a rigorous testbed for model robustness. On the visual side, variations in head poses, illumination, and image quality make the dataset even more realistic and challenging. The mixed data version consists of 56 hours of training data, 3 hours of validation data, and 1.5 hours of test data.

For a fair comparison with existing AVSS methods, we adopted preprocessing pipelines consistent with prior work. For the visual modality, we first used FFmpeg to resample all videos to 25 FPS.

A widely used face detection network (Zhang et al., 2016) was then applied to localize and crop the lip region, which was resized into a $96 \times 96$ grayscale sequence. For the auditory modality, FFmpeg was used to extract audio tracks from the original videos and resample them into 16 kHz mono signals. During dataset construction, we followed the standard paradigm of mixing clean audio signals from two distinct speakers to generate training samples. While this study focuses on two-speaker mixtures, the data generation process is inherently scalable to scenarios involving three or more speakers.

## D  AVSS OBJECTIVE FUNCTION

We adopt time-domain SI-SNR and frequency-domain SI-SNR (Hershey et al., 2016) as the objective functions to be minimized when training Dolphin. The time-domain SI-SNR is defined as

$$\text{SI-SNR}_t(\mathbf{S}, \hat{\mathbf{S}}) = 10 \log_{10} \frac{\|\omega\,\mathbf{S}\|_2^2}{\left\|\hat{\mathbf{S}} - \omega\,\mathbf{S}\right\|_2^2}, \tag{21}$$

$$\omega = \frac{\hat{\mathbf{S}}^\top \mathbf{S}}{\mathbf{S}^\top \mathbf{S}}, \tag{22}$$

where $\mathbf{S}$ and $\hat{\mathbf{S}}$ denote the ground-truth and the estimated speech signals, respectively.

For the frequency-domain SI-SNR, we start from the separator's decoder-layer output $\mathbf{D}_3 \in \mathbb{R}^{N_a \times (T_a/8)}$. We first upsample along the temporal axis to align $T_a/8$ to $T_a$, then apply a convolution followed by a ReLU activation, dot-multiply with $\mathbf{X}$, and finally pass the result through the audio decoder to obtain the time-domain signal $\hat{\mathbf{S}}_3 \in \mathbb{R}^{1 \times L_a}$. Let $\mathcal{F}(\cdot)$ denote the STFT operator. The magnitude spectra of the ground-truth and estimated speech are given by

$$\mathbf{M} = |\mathcal{F}(S)|, \quad \hat{\mathbf{M}} = |\mathcal{F}(\hat{\mathbf{S}}_3)|, \tag{23}$$

where $\mathbf{M}, \hat{\mathbf{M}} \in \mathbb{R}^{F \times T}$ are the magnitude spectra, $F$ is the number of frequency bins, and $T$ is the number of frames. Vectorizing the magnitude spectra yields

$$\mathbf{m} = \text{vec}(\mathbf{M}) \in \mathbb{R}^{FT}, \qquad \hat{\mathbf{m}} = \text{vec}(\hat{\mathbf{M}}) \in \mathbb{R}^{FT}. \tag{24}$$

We define the frequency-domain projection coefficient as

$$\omega_f = \frac{\hat{\mathbf{m}}^\top \mathbf{m}}{\mathbf{m}^\top \mathbf{m}}, \tag{25}$$

and the frequency-domain SI-SNR is

$$\text{SI-SNR}_f(\mathbf{S}, \hat{\mathbf{S}}_3) = 10 \log_{10} \frac{\|\omega_f\,\mathbf{m}\|_2^2}{\|\hat{\mathbf{m}} - \omega_f\,\mathbf{m}\|_2^2}. \tag{26}$$

The overall loss is defined as

$$\mathcal{L}(\mathbf{S}, \hat{\mathbf{S}}_3) = (1 - \lambda)\,\text{SI-SNR}_t(\mathbf{S}, \hat{\mathbf{S}}) + \lambda\,\text{SI-SNR}_f(\mathbf{S}, \hat{\mathbf{S}}_3), \tag{27}$$

where, following a strategy similar to Sepreformer (Shin et al., 2024), we adopt epoch-dependent weighting:

$$\lambda = \begin{cases} 0.4, & \text{if epoch} \leq 80, \\ 0.4 \times 0.8^{\lfloor (\text{epoch} - 80)/5 \rfloor}, & \text{if epoch} > 80. \end{cases} \tag{28}$$

## E  MODEL HYPERPARAMETER CONFIGURATIONS

To ensure reproducibility of experimental results and to facilitate future extensions, we provide detailed descriptions of the structural configurations and key hyperparameters of each sub-module. Unless otherwise specified, all convolutions are one-dimensional in the temporal domain. The notations are defined as follows: convolution kernel size $k$, stride $s$, number of attention heads $H$, per-head dimension $d_h$, feed-forward channel size $d_{\text{FFN}}$, and hidden dimension $d_{\text{hid}}$.

**Visual Encoder.** We employ a pretrained visual encoder and utilize both reconstruction tokens and semantic tokens as visual inputs. These tokens are fused through element-wise summation to obtain the visual feature vector, with dimensionality $d_v = 3872$. This representation serves as the input to subsequent stages of visual feature integration and cross-modal modeling.

**Audio Encoder and Decoder.** The audio encoder applies a one-dimensional convolution ($k = 16, \ s = 4$) to map the raw waveform into a compact latent space, producing a 256-dimensional audio representation. The audio decoder adopts a transposed one-dimensional convolution, configured symmetrically to the encoder ($k = 16, \ s = 4$), in order to progressively upsample and reconstruct the latent representation back to the waveform domain.

**Visual Feature Fusion Network.** The visual feature fusion network in AVF module serves as a preprocessing stage for the input video features. We reuse the TDANet module, consistent with RTFSNet, based on the official open-source implementation. The main configuration includes hidden dimension $d_{\text{hid}} = 64$, convolution kernel $k = 3$, upsampling depth of 4, Batch Normalization layers, an 8-head multi-head self-attention (MHSA), and a feed-forward network with channel size $d_{\text{FFN}} = 128$.

**AVF Module.** For cross-modal alignment and fusion, this module is constructed with hidden dimension $d_{\text{hid}} = 128$, kernel size $k = 1$ ($1 \times 1$ convolution), an upsampling depth of 4, and Layer Normalization as the normalization strategy.

**Separator.** The backbone adopts a hierarchical encoder–decoder structure with 4 downsampling layers. In each CSA block, MHSA is configured with $H = 8$ attention heads and per-head dimension $d_h = 128$. In the MSA blocks, we employ a convolution kernel size $k = 3$ and hidden dimension $d_{\text{hid}} = 128$.

## F    DETAILS OF DIFFERENT PRETRAINED VIDEO ENCODER

**3D ResNet-18:** This model first applies a 3D convolution to extract temporal and spatial features, followed by using a standard ResNet-18 (He et al., 2016) to obtain visual representations for each frame. The majority of its computational cost comes from the ResNet-18 backbone.

**AE:** A lightweight 2D autoencoder that represents a simpler and more computationally efficient approach. The model processes each video frame independently, and its architecture is consistent with that of AVLiT-8 (Martel et al., 2023).

## G    ABLATION STUDY: EFFECT OF THE VECTOR QUANTIZATION MODULE IN THE DP-LIPCODER

To assess the contribution of the vector quantization (VQ) module in our semantic path, we conduct an ablation experiment by removing VQ entirely from the visual branch. Specifically, we eliminate the quantizer and its associated commitment loss while retaining the reconstruction and distillation objectives. The modified model is retrained on the same datasets as the original DP-LipCoder, and the separator is trained from scratch with the visual encoder kept frozen during separation training.

Table 7 summarizes the results. Removing VQ leads to a consistent degradation across all metrics, most notably a drop of approximately 0.5 dB in SI-SNRi, demonstrating that VQ plays an important role in improving semantic alignment and separation quality.

Table 7: Ablation results for the VQ module on LRS2.

| Method | SI-SNRi ↑ | SDRi ↑ | PESQ ↑ |
|---|---|---|---|
| Without VQ | 16.3 | 16.5 | 3.27 |
| With VQ | 16.8 | 16.9 | 3.29 |

We attribute this improvement to VQ's ability to compress continuous features into discrete semantic anchors, which enhances temporal alignment and stabilizes the distillation process against distribution drift. Furthermore, the discrete bottleneck serves as a regularizer that suppresses irrelevant

variations, thereby improving robustness to noise and generalization with negligible computational overhead.

## H  MULTI-SPEAKER PERFORMANCE

Table 8: Performance comparison of different methods on LRS2 multi-speaker separation tasks.

| Methods | LRS2-2Mix | | LRS2-3Mix | | LRS2-4Mix | |
|---|---|---|---|---|---|---|
| | SI-SNRi ↑ | SDRi ↑ | SI-SNRi ↑ | SDRi ↑ | SI-SNRi ↑ | SDRi ↑ |
| AV-ConvTasNet (Wu et al., 2019) | 12.5 | 12.8 | 8.2 | 8.8 | 4.1 | 4.6 |
| AVLIT-8 (Martel et al., 2023) | 12.8 | 13.1 | 9.4 | 9.9 | 5.0 | 5.7 |
| CTCNet (Li et al., 2024a) | 14.3 | 14.6 | 10.3 | 10.8 | 6.3 | 6.9 |
| IIANet (Li et al., 2024b) | 16.0 | 16.2 | 12.6 | 13.1 | 7.8 | 8.3 |
| Dolphin (*ours*) | **16.8** | **16.9** | **13.1** | **13.3** | **9.7** | **9.9** |

To ensure a fair comparison, we followed the experimental protocol adopted in IIANet (Li et al., 2024b). Specifically, the proposed model was trained and evaluated on the LRS2-3Mix and LRS2-4Mix datasets. For each dataset containing $N$ speakers, a dedicated model was trained to handle mixtures with that specific number of speakers. During evaluation, we employed the same iterative inference strategy as in IIANet: for each target speaker in a mixture, the model sequentially extracted the corresponding audio stream.

We compared our approach against several baseline methods, including AV-ConvTasNet (Wu et al., 2019), AVLIT-8 (Martel et al., 2023), CTCNet (Li et al., 2024a), and IIANet (Li et al., 2024b). To ensure fairness, all baseline results were directly taken from IIANet. The quantitative comparison is reported in Table 8. The results clearly indicate that our proposed method consistently outperforms all existing baselines across all evaluation datasets in terms of multiple performance metrics. Specifically, our proposed Dolphin achieves the largest improvement over the state-of-the-art on LRS2-4Mix, the dataset with the greatest number of speakers, highlighting its superior separation performance in complex real-world scenarios.

## I  ADDITIONAL EXPERIMENTS UNDER DIVERSE ACOUSTIC CONDITIONS

Although the datasets we used for training and evaluation(e.g., LRS2, VoxCeleb2) mostly originate from real-world sources(e.g. BBC broadcasts, YouTube videos), we additionally constructed a set of challenging test samples to further evaluate the robustness of our model. Specifically, we adopted VoxCeleb Chung et al. (2018) as the base and introduced environmental noise(from FSD50K Fonseca et al. (2022)), music noise(from FMA Defferrard et al. (2017)) and multiple interfering speakers. Finally, we constructed four challenging test scenarios based on the VoxCeleb2 dataset:

- **Scenario 1 (Environmental Noise):** 1 speaker + environmental noise from FSD50K.

- **Scenario 2 (Music Noise):** 1 speaker + music noise from FMA.

- **Scenario 3 (Environmental Noise + 2 Speakers):** 1 speaker + environmental noise + 2 additional interfering speakers (3 speakers total).

- **Scenario 4 (Music Noise + 2 Speakers):** 1 speaker + music noise + 2 additional interfering speakers (3 speakers total).

Each scenario contains 1000 test samples. We evaluated Dolphin along with two representative baselines (IIANet and AV-Mossformer2) across all scenarios. The results, summarized in Table 9, show that Dolphin consistently outperforms the baselines on all metrics (SI-SNRi, SDRi, PESQ), demonstrating greater robustness and separation capability, particularly in the most challenging conditions involving three speakers and additional noise. These experiments complement the main results reported in Section 5 and provide additional empirical support for the effectiveness of Dolphin under realistic and more adverse acoustic environments.

Table 9: Performance under four noisy scenarios.

(a) Scenario 1: 1 speaker + environmental noise

| Methods | SI-SNRi ↑ | SDRi ↑ | PESQ ↑ |
|---|---|---|---|
| IIANet | 6.69 | 9.71 | 2.07 |
| AV-Mossformer2 | 8.33 | 9.01 | 2.13 |
| Dolphin (ours) | 11.41 | 12.36 | 2.43 |

(b) Scenario 2: 1 speaker + music noise

| Methods | SI-SNRi ↑ | SDRi ↑ | PESQ ↑ |
|---|---|---|---|
| IIANet | 2.04 | 5.92 | 1.64 |
| AV-Mossformer2 | 6.68 | 7.53 | 1.76 |
| Dolphin (ours) | 7.49 | 8.52 | 1.85 |

(c) Scenario 3: 1 speaker + environmental noise + 2 interfering speakers

| Methods | SI-SNRi ↑ | SDRi ↑ | PESQ ↑ |
|---|---|---|---|
| IIANet | 1.09 | 4.64 | 1.44 |
| AV-Mossformer2 | 3.33 | 4.71 | 1.49 |
| Dolphin (ours) | 5.37 | 6.56 | 1.57 |

(d) Scenario 4: 1 speaker + music noise + 2 interfering speakers

| Methods | SI-SNRi ↑ | SDRi ↑ | PESQ ↑ |
|---|---|---|---|
| IIANet | 0.54 | 3.63 | 1.36 |
| AV-Mossformer2 | 3.90 | 4.56 | 1.41 |
| Dolphin (ours) | 4.11 | 5.33 | 1.46 |

# J  SUBJECTIVE HUMAN EVALUATION ON REAL OVERLAPPING SPEECH

A large body of prior audio-visual speech separation (AVSS) research is based on datasets such as LRS2, LRS3, and VoxCeleb2, where overlapping mixtures are artificially constructed. While these datasets have become standard benchmarks and enable reproducible supervised training, the resulting mixtures differ from real-world overlapping speech, where speakers naturally adjust their vocal effort in response to background noise (an effect known as the Lombard effect). To better assess practical performance under realistic acoustic conditions, we further conducted a Mean Opinion Score (MOS) evaluation on challenging in-the-wild overlapping speech.

We collected 18 short clips (10-20 seconds) from publicly available debate videos that contain natural overlap speech, background noise, and room reverberation. A total of 50 human listeners participated in the study. Each listener rated the outputs of IIANet, AV-Mossformer2, and our Dolphin using the standard 5-point MOS scale (1 = poor, 5 = excellent). The evaluation was performed in a double-blind manner with randomized audio order.

Table 10 summarizes the average MOS results across all clips. Dolphin achieves substantially higher scores across all criteria, demonstrating clearer separation and fewer audible artifacts compared to competing baselines. These subjective findings are consistent with qualitative examples on our demo page.

# K  VISUAL RESULTS FOR SPEECH SEPARATION

We conduct a visual analysis of the separation performance of three audio-visual speaker separation (AVSS) approaches: Dolphin, IIANet, and AV-Mossformer2. The models are trained on the LRS2 dataset, and representative results are shown in Figure 9. To evaluate these methods, we randomly select four audio mixtures from LRS2 as test samples.

Table 10: Mean Opinion Score (MOS) results on real overlapping-speech recordings.

| Method | MOS |
|---|---|
| IIANet | 2.24 ± 0.15 |
| AV-Mossformer2 | 2.85 ± 0.27 |
| Dolphin (ours) | 3.86 ± 0.25 |

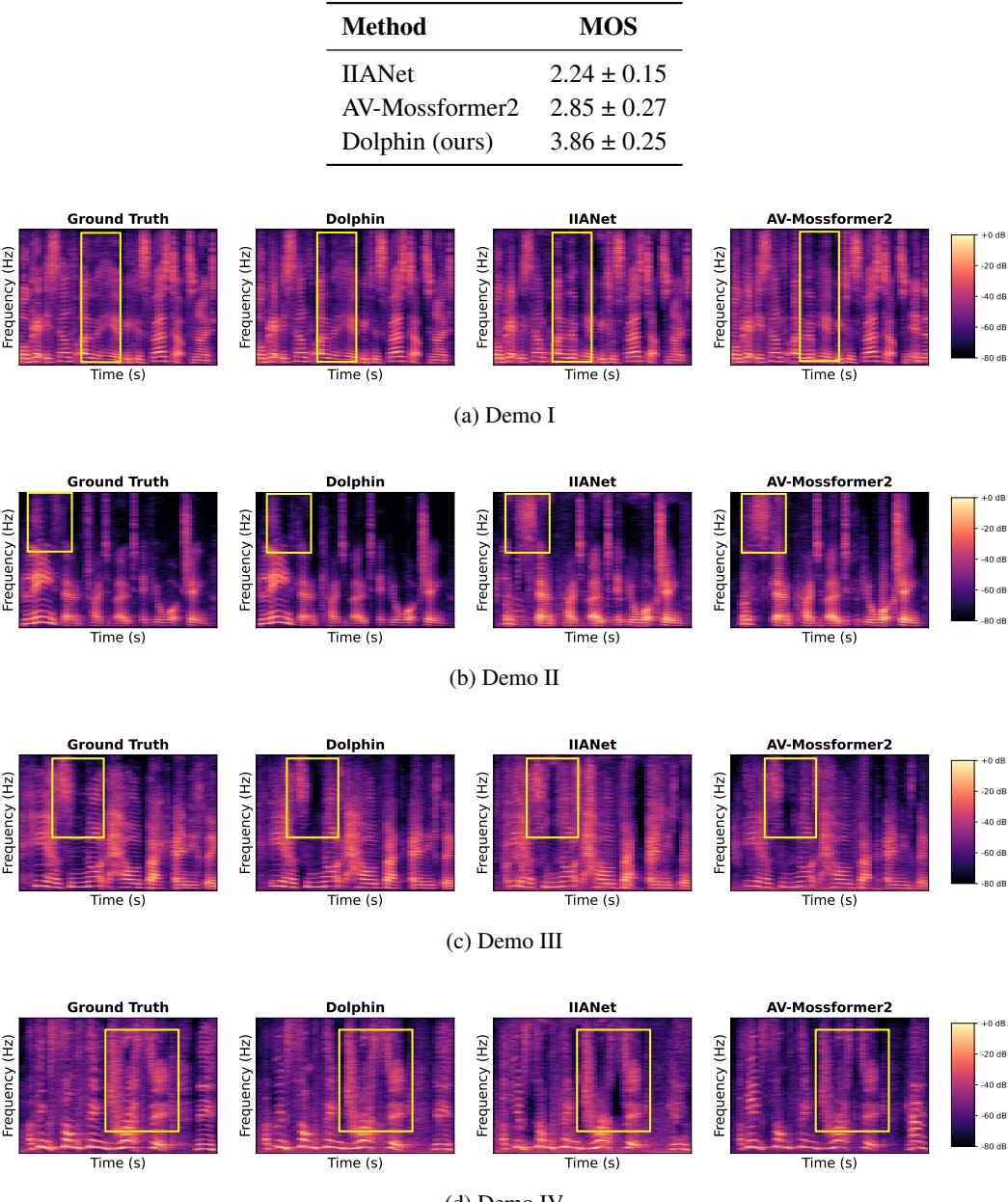

(a) Demo I

(b) Demo II

(c) Demo III

(d) Demo IV

Figure 9: Visualization of separation results obtained by different methods on four examples.

In the first sample (Figure 9a), Dolphin accurately reconstructs the main harmonic structure of speech, while also producing a cleaner spectrogram with noticeably less background noise. By contrast, the output of IIANet contains more noise artifacts.

In the second sample (Figure 9b), Dolphin again achieves the best performance, with its spectrogram most closely resembling the ground-truth speech. Although AV-Mossformer2 and IIANet are able to separate speech, both methods fail to preserve certain low-frequency details.

In the third sample (Figure 9c), Dolphin remains the most effective, retaining the largest proportion of speech components. Results from AV-Mossformer2 and IIANet exhibit significant residual noise, especially in the high-frequency bands.

In the fourth sample (Figure 9d), Dolphin once more demonstrates clear superiority by recovering speech with a complete and well-defined harmonic structure. In comparison, both AV-Mossformer2 and IIANet struggle to reconstruct a coherent harmonic pattern.

Across all four test cases, Dolphin consistently delivers the best separation results. Its outputs are the most similar to the ground-truth spectrograms, with the least noise and the most complete harmonic structures.

## L  IMPACT OF THE NUMBER OF SEPARATOR ITERATIONS

Table 11: Impact of the number of separator iterations on separation performance and efficiency.

| Methods | SI-SNRi↑ | SDRi↑ | PESQ↑ | Params (MB)↓ | MACs (G/s)↓ | Latency (ms)↓ | |
|---|---|---|---|---|---|---|---|
| | | | | | | CPU | GPU |
| AV-TDANet-1 | 6.4 | 7.6 | 2.32 | 3.89 | 2.99 | 676.21 | 15.80 |
| AV-TDANet-8 | 12.4 | 12.8 | 2.87 | 3.89 | 7.04 | 1185.80 | 44.67 |
| AV-TDANet-16 | 12.8 | 13.2 | 2.89 | 3.89 | 11.68 | 2981.78 | 78.30 |
| **Dolphin** | **16.8** | **16.9** | **3.29** | **7.00** | **10.89** | **2117.96** | **33.24** |

To validate the effectiveness of the proposed single-iteration design with GLA block, we incorporate the same AVF module as in Dolphin into the baseline audio-only speech separation network TDANet, resulting in AV-TDANet. We compared AV-TDANet with 8 and 16 iterations, as specified in the original TDANet paper, against our single-iteration Dolphin. For a fair ablation, we additionally evaluated AV-TDANet with one iteration. The construction of AV-TDANet only differs in replacing the Dolphin separator with TDANet. As shown in Table 11, increasing the number of iterations substantially increases the computational burden: MACs grow from 2.99 G/s at 1 iteration to 7.04 G/s at 8 iterations ($2.35\times$), and further to 11.68 G/s at 16 iterations ($3.9\times$ compared to 1 iteration). In contrast, Dolphin achieves comparable results with much lower overhead, enabled by the GLA module, which captures both global and local dependencies and ensures a favorable efficiency–performance trade-off.

## M  ANALYSIS OF ENCODER AND DECODER ARCHITECTURE

Table 12: Ablation study on different allocations of GLA modules between encoder and decoder (total number = 5).

| Encoder Layers | Decoder Layers | SI-SNRi↑ | SDRi↑ | PESQ↑ | Params (MB)↓ | MACs (G/s)↓ |
|---|---|---|---|---|---|---|
| 1 | 4 | 16.7 | 16.8 | 3.29 | 7.00 | 10.89 |
| **2** | **3** | **16.8** | **16.9** | **3.29** | **7.00** | **10.89** |
| 3 | 2 | 16.5 | 16.6 | 3.25 | 7.00 | 10.89 |
| 4 | 1 | 16.1 | 16.2 | 3.24 | 7.00 | 10.89 |

To systematically evaluate the optimal depth configuration of GLA modules within the encoder and decoder, we fixed the total number of GLA modules to 5 and designed several allocation strategies for testing. The primary objective of this experiment is to analyze the relative contributions of the encoding and decoding stages to overall performance improvement. The results are presented in Table 12. It can be observed that the configuration with 2 GLA modules in the encoder and 3 in the decoder achieves the best performance across all evaluation metrics. In contrast, other configurations fail to reach the same performance level. This finding suggests that a relatively deeper decoder plays a critical role in accurately reconstructing high-quality target speech from high-level semantic representations, and further underscores the importance of carefully balancing modeling capacity between the encoder and decoder.

## N  ABLATION STUDY: OUTPUT FEATURE FORMULATION OF THE SEPARATOR

In order to investigate the optimal representation of the separator output features, we conducted an ablation study. Mainstream speech separation approaches are typically mask-based (Wu et al.,

2019; Li et al., 2024a;b; Pegg et al., 2023; Zhao et al., 2025; Sang et al., 2025), where the separator generates a feature mask that is applied to the encoder's mixture representation through element-wise multiplication to estimate the target speaker representation. In contrast, we adopt a direct mapping strategy: instead of employing a masking mechanism, the decoder directly regresses the embedding of the target speaker. We hypothesize that directly predicting the target representation can mitigate potential nonlinear distortions or information loss introduced by the masking operation (Wang et al., 2023).

Table 13: Comparison of direct mapping and masking strategies on the LRS2 dataset.

| Methods | SI-SNRi ↑ | SDRi ↑ | PESQ ↑ |
|---------|-----------|--------|--------|
| Mapping | 16.8 | 16.9 | 3.29 |
| Mask | 16.3 | 16.4 | 3.20 |

We evaluated both approaches (direct mapping vs. masking) on the LRS2 dataset, and the results are summarized in Table 13. The direct mapping strategy consistently outperforms the masking-based approach across all objective metrics. Specifically, our method achieves a $0.5$ dB improvement in both SI-SNRi and SDRi, while also obtaining an additional $0.09$ in PESQ. These results strongly demonstrate the effectiveness of direct mapping, suggesting that regressing the target representation directly from the mixture provides a superior paradigm. Consequently, our final model adopts the direct mapping scheme.

## O  ABLATION STUDY: EFFECT OF FUSION LOCATION

To systematically evaluate the impact of fusion position between visual and auditory features on speech separation performance, we conducted comparative experiments at four candidate positions $\mathbf{F}_0 \sim \mathbf{F}_3$, as illustrated in Figure 3. Specifically, $\mathbf{F}_0$ denotes the earliest fusion at the beginning of the acoustic encoding process, while $\mathbf{F}_1 \sim \mathbf{F}_3$ progressively inject visual features into deeper stages of speech representations. Apart from the fusion position, all training and inference configurations were kept identical to ensure a fair comparison.

Table 14: Comparison of different fusion positions on the LRS2 dataset.

| Fusion Position | SI-SNRi | SDRi | PESQ |
|-----------------|---------|------|------|
| $\mathbf{F}_0$ | 16.8 | 16.9 | 3.29 |
| $\mathbf{F}_1$ | 16.5 | 16.6 | 3.22 |
| $\mathbf{F}_2$ | 16.3 | 16.4 | 3.21 |
| $\mathbf{F}_3$ | 16.2 | 16.3 | 3.21 |

Table 14 reports the separation performance on the LRS2 dataset. As shown, the earliest fusion $\mathbf{F}_0$ achieves the best results across all three metrics: compared to the latest fusion $\mathbf{F}_3$, it yields improvements of +0.6 dB in SI-SNRi, +0.6 dB in SDRi, and +0.08 in PESQ. Moreover, as the fusion position shifts from $\mathbf{F}_0$ to $\mathbf{F}_3$, the performance consistently declines. This observation aligns with findings in neuroscience regarding cross-modal interactions (Kuo et al., 2022).

## P  LLM USAGE

In preparing this manuscript, the authors used a large language model (LLM) solely to assist with polishing the wording and improving the clarity of English expression. The LLM did not contribute to the research design, data collection, data analysis, or the generation of scientific ideas. Its role was limited to helping refine sentence structure, grammar, and flow of the written text. All substantive content, methods, and conclusions are entirely the authors' own work.

