# OpenReview forum: "Efficient Audio-Visual Speech Separation with Discrete Lip Semantics and Multi-Scale Global-Local Attention"
_ICLR.cc/2026/Conference — ICLR 2026 Poster_

### Official Review · Reviewer_mwcV · 2025-10-30

**Soundness:** 3
**Presentation:** 4
**Contribution:** 4
**Rating:** 6
**Confidence:** 3

**Summary:**

The paper introduces Dolphin, a compact and efficient audio-visual speech separation framework that integrates visual motion cues with attention-based processing to isolate target voices. It employs DP-LipCoder, a lightweight dual-path video encoder that learns discrete visual representations aligned with speech, and a single-iteration separator with global–local attention for efficient feature modeling. Through its optimized design, Dolphin achieves superior separation quality with over 50% fewer parameters, 2.4× lower computational cost, and 6× faster inference, making it well-suited for real-time and edge deployment.

**Strengths:**

For the AVSS task, we designed a highly efficient model that outperforms existing approaches in key metrics such as SI-SNRi and SDRi, while achieving a 50% reduction in parameters and 2.4× lower computational cost, effectively balancing the trade-off between performance and efficiency.

Also, throughout the paper, reasonable architectural designs such as the single-pass GLA mechanism and the DP-LipCoder, which quantizes lip movements into discrete tokens, were employed to reduce computational cost while maintaining high performance. The effectiveness of these architectural innovations is well supported by the experimental results.

**Weaknesses:**

There is a lack of ablation studies for individual components. In particular, an additional experiment comparing the performance without using VQ, which plays a crucial role in this work, would be valuable.

Moreover, the paper lacks sufficient analysis of the individual contributions of global and local attention within the GLA blocks, as well as comparisons with other audio-visual fusion strategies

It is also unclear whether Dolphin can maintain its performance without a pretrained teacher model such as AV-HuBERT.

* Minor weaknesses
    * Line 234: The link for Figure 3 should include the letter “F.”
    * For better experimental transparency, it would be helpful to indicate which results in Table 3 were taken from the original papers.

**Questions:**

* It is unclear whether the encoders in the Semantic Path and Reconstruction Path shown in Figure 2 are trained separately or if there is parameter sharing between them. A more detailed explanation would be helpful.

* For the results reported in Table 2, it is not specified whether the performance was measured after retraining the model with DP-LipCoder or simply by replacing the visual encoder without additional training. Clarification on this point is needed.

* Including a discussion on the limitations of this work would make the paper more comprehensive and could provide valuable guidance for future research.

---

> ### Author Response · Authors · 2025-11-20
>
> **Q1: In particular, an additional experiment comparing the performance without using VQ, which plays a crucial role in this work, would be valuable.**
>
> **A1:** We thank the reviewer for highlighting the importance of evaluating the necessity of VQ. In response, we conducted an ablation study removing the vector quantizer (VQ) from the semantic path. Specifically, we completely removed VQ from visual branch and retrained it (The objective retains the reconstruction and distillation terms, while removing only the VQ commitment term), and then trained the separator from scratch on LRS2, keeping the visual encoder frozen during separation. The results, summarized in the table below, show that VQ provides an improvement of approximately 0.5 dB in SI-SNRi.
>
> | Methods | SI-SNRi ↑ | SDRi ↑ | PESQ ↑ |
> | :--- | :---: | :---: | :---: |
> | without VQ | 16.3 | 16.5 | 3.27 |
> | **with VQ** | **16.8** | **16.9** | **3.29** |
>
> We attribute this improvement to VQ’s ability to compress continuous features into discrete semantic anchors, which enhances temporal alignment and stabilizes the distillation process against distribution drift. Furthermore, the discrete bottleneck serves as a regularizer that suppresses irrelevant variations, thereby improving robustness to noise and generalization with negligible computational overhead. We have included the results in the revised manuscript appendix.
>
> **Q2: The paper lacks sufficient analysis of the individual contributions of global and local attention within the GLA blocks.**
>
> **A2:** We would like to clarify that systematic ablation studies of the Global Attention(GA) and Local Attention (LA) modules are already presented in Section 5.3 and Table 5 of the manuscript, quantifying their individual contributions and complementary effects. We have further added some analysis in the revised manuscript. Specifically, Table 5 evaluates four variants of the model:
>
> 1. Baseline model: Both GA and LA modules are removed, replaced with simple depthwise convolutions to preserve the basic network structure.
> 2. LA only: Only the LA module is retained; GA is removed.
> 3. GA only: Only the GA module is retained; LA is removed.
> 4. Dolphin: Both GA and LA modules are included.
>
> The results clearly demonstrate the indispensability of both modules. First, the baseline model without any specialized attention performs poorly, with SI-SNRi of only 10.4 dB. Introducing the LA module alone increases SI-SNRi to 15.6 dB, yielding a gain of 5.2 dB. This confirms that the LA module, through its Heat Diffusion Attention (HDA) mechanism, effectively smooths features in the frequency domain, suppresses noise, and accurately models local structures of the audio signal. Similarly, introducing the GA module alone raises SI-SNRi to 15.9 dB, a gain of 5.5 dB, highlighting the GA module’s ability to efficiently capture global context and long-range dependencies via Coarse-grained Self-Attention (CSA) applied on downsampled, low-resolution feature maps.
>
> Crucially, when both GA and LA modules are combined (i.e., our Dolphin model), SI-SNRi reaches 16.8 dB. This not only far exceeds the baseline but also improves over the “GA only” and “LA only” variants by 0.9 dB and 1.2 dB, respectively. GA and LA are not functionally redundant; rather, they model complementary levels of audio features: GA captures global contextual information, while LA models local structural details. The effective integration of global and local information enables our separator to achieve or even surpass the performance of prior models that rely on multiple iterative passes, thereby reaching an excellent balance between efficiency and separation.

---

> > ### Author Response · Authors · 2025-11-20
> >
> > **Q3: Comparisons with other audio-visual fusion strategies.**
> >
> > **A3:** Following the reviewer’s suggestion, we conducted a series of new experiments comparing our method with three widely used fusion strategies: simple addition (`add`), channel-wise concatenation (`concat`), and the standard `cross-attention` mechanism. We carefully implemented these baselines to ensure fair and rigorous comparisons. We have included the results and analysis in the appendix of the revised version.
> >
> > The implementation details are as follows:
> > * **Addition fusion:** We first process the visual and audio streams separately using independent convolutional layers to extract unimodal features. The visual feature extractor is followed by a BatchNorm layer, while the audio stream uses a GroupNorm layer. The visual feature map is then spatially upsampled to match the audio feature’s temporal dimension. Finally, the two feature tensors are fused via element-wise addition.
> > * **Concatenation fusion:** This strategy shares the same initial feature extraction and upsampling steps as the `add` method. However, the visual and audio feature tensors are concatenated along the channel dimension. The concatenated tensor is then processed by an additional convolutional layer followed by GroupNorm to integrate information and reduce the channel dimensionality.
> > * **Cross-attention fusion:** We also implemented a Transformer-based cross-modal attention module. In this configuration, audio features serve as queries, while visual features act as both keys and values. This allows the model to dynamically weight and aggregate relevant visual information for each audio time step, enabling a more sophisticated fusion mechanism.
> >
> > | Method | SI-SNRi ↑ | SDRi ↑ | PESQ ↑ |
> > | :--- | :---: | :---: | :---: |
> > | add | 16.0 | 16.1 | 3.25 |
> > | concat | 16.3 | 16.4 | 3.26 |
> > | cross-attention | 14.0 | 14.4 | 2.91 |
> > | **ours** | **16.8** | **16.9** | **3.29** |
> >
> > The experimental results are summarized in the table above, highlighting the superiority of our method. Several key observations can be made: First, the two simple fusion strategies, `add` and `concat`, achieve reasonable performance, confirming the benefit of incorporating visual information. Among them, `concat` slightly outperforms `add`, indicating that preserving the unique characteristics of each modality before fusion offers a modest advantage over direct element-wise addition. Second, more importantly, our proposed method significantly outperforms both simple baselines across all evaluation metrics. For example, it achieves an SI-SNRi of 16.8 dB, improving by 0.5 dB over the stronger `concat` baseline. This demonstrates that our method captures complex nonlinear cross-modal correlations more effectively than simple algebraic operations. Third, it is noteworthy that the `cross-attention` mechanism, despite its higher complexity, performs the worst, even below the `add` and `concat` baselines. We hypothesize that in this task setting, a standard cross-attention implementation may struggle to learn meaningful audio-visual correspondences without more specialized architectural design or extensive pretraining. It may be prone to overfitting or focusing on spurious correlations, leading to performance degradation. In contrast, our method is specifically designed to model the correlation between audio and visual cues, thereby achieving SOTA performance.

---

> > > ### Author Response · Authors · 2025-11-20
> > >
> > > **Q4: It is also unclear whether Dolphin can maintain its performance without a pretrained teacher model such as AV-HuBERT.**
> > >
> > > **A4:** We further conducted a new ablation study on LRS2 to evaluate the necessity of the teacher model. Specifically, we removed the semantic path that relies on AV-HuBERT distillation, retaining only the reconstruction path. That is, we pretrained a lightweight visual autoencoder purely via video reconstruction, and used its encoded features as input to Dolphin. The results, summarized in the table below, show that removing the semantic path and AV-HuBERT distillation leads to only a ~0.7 dB drop in SI-SNRi, while still significantly outperforming the AE baseline that relies solely on frame-wise reconstruction (SI-SNRi = 15.2 dB on LRS2, Table 1).
> > >
> > > | Methods | SI-SNRi ↑ | SDRi ↑ | PESQ ↑ |
> > > | :--- | :---: | :---: | :---: |
> > > | Only reconstruction path | 16.1 | 16.2 | 3.22 |
> > > | Only semantic path | 16.3 | 16.4 | 3.24 |
> > > | **DP-LipCoder** | **16.8** | **16.9** | **3.29** |
> > >
> > > These results indicate that AV-HuBERT is used only during pretraining to provide high-level audio-visual alignment supervision for the semantic path, yielding roughly 0.7 dB performance improvement. It is not a prerequisite for Dolphin’s feasibility. From an engineering perspective, the one-time teacher distillation during pretraining is manageable and does not affect the model complexity or inference latency at deployment, which involves only the frozen DP-LipCoder and lightweight GLA separator; their parameter count and MACs are fully reported in Tables 1 and 4.
> > >
> > > In summary, Dolphin can achieve comparable separation performance even without a teacher model (using only the reconstruction path). Incorporating AV-HuBERT distillation is an optional enhancement that further improves performance. Therefore, the reviewer’s concern is empirically addressed.
> > >
> > > **Q5: Line 234: The link for Figure 3 should include the letter “F.”**
> > >
> > > **A5:** It has been corrected. Thanks.
> > >
> > > **Q6: For better experimental transparency, it would be helpful to indicate which results in Table 3 were taken from the original papers.**
> > >
> > > **A6:** Except for AV-Mossformer2 and Swift-Net, the results in Table 3 for all other methods including AV-ConvTasNet, VisualVoice, AVLiT-8, CTCNet, RTFS-Net, and IIANet are directly taken from their original papers [1,2,3,4,5,6]. We did not reimplement or retrain these models to ensure strict consistency with the published results.
> > >
> > > For AV-Mossformer2 and Swift-Net, the results were reproduced using the authors’ publicly available official codebases under our unified data preprocessing and experimental settings. Specifically, AV-Mossformer2 was implemented using the official [ClearerVoice-Studio repository](https://github.com/modelscope/ClearerVoice-Studio/tree/main/train/target_speaker_extraction), and Swift-Net used the authors’ official [repository](https://github.com/JusperLee/Swift-Net).
> > >
> > > We have updated Table 3 and the corresponding text in the revised manuscript to reflect this clarification.

---

> > > > ### Author Response · Authors · 2025-11-20
> > > >
> > > > **Q7: It is unclear whether the encoders in the Semantic Path and Reconstruction Path shown in Figure 2 are trained separately or if there is parameter sharing between them. A more detailed explanation would be helpful.**
> > > >
> > > > **A7:** In Figure 2, the semantic path and reconstruction path of DP-LipCoder adopt the same encoder architecture; however, the encoder parameters are not shared between the two paths. We will explicitly clarify in Section 3.2 of the revised manuscript that the semantic and reconstruction path encoders are trained separately, to eliminate any potential ambiguity arising from the current illustration.
> > > >
> > > > **Q8: For the results reported in Table 2, it is not specified whether the performance was measured after retraining the model with DP-LipCoder or simply by replacing the visual encoder without additional training. Clarification on this point is needed.**
> > > >
> > > > **A8:** All results in Table 2 that use DP-LipCoder were obtained by retraining the separation models from scratch after replacing the visual encoder. It is important to note that this procedure is consistent with the original papers: in the prior work, the audio separation network is also trained from scratch on a fixed, pretrained visual backbone. We simply follow the same paradigm, replacing the original visual backbone with DP-LipCoder. In the final version, we will explicitly state in the caption of Table 2 that all results using DP-LipCoder were obtained by retraining the AVSS model from scratch with the new frozen visual encoder.
> > > >
> > > > **Q9: Including a discussion on the limitations of this work would make the paper more comprehensive and could provide valuable guidance for future research.**
> > > >
> > > > **A9:** In the revised manuscript, we will explicitly discuss the main limitations as follows:
> > > >
> > > > **Limitations and Future Work.** While Dolphin balances efficiency and performance, several avenues remain for exploration. One limitation lies in the reliance on relatively clean and synchronized lip videos; future work should enhance robustness against large head pose variations, occlusions, and extreme lighting conditions. Additionally, although our model significantly reduces computational costs compared to prior AVSS systems, deployment on extremely resource-constrained edge devices remains a challenge. We plan to investigate techniques like quantization or pruning to bridge this gap. Finally, while mapping lip motion to discrete tokens improves efficiency, it may discard fine-grained articulatory cues. Exploring hierarchical codebooks or mixed discrete–continuous representations offers a promising direction to recapture this subtle visual information.
> > > >
> > > > ### References
> > > >
> > > > [1] Kai Li, Fenghua Xie, Hang Chen, Kexin Yuan, Xiaolin Hu. An Audio-Visual Speech Separation Model Inspired by Cortico-Thalamo-Cortical Circuits (CTCNet). arXiv preprint arXiv:2212.10744, 2022.
> > > >
> > > > [2] Jian Wu, Yong Xu, Shi-Xiong Zhang, Lian-Wu Chen, Meng Yu, Lei Xie, Dong Yu. Time Domain Audio-Visual Speech Separation (AV-ConvTasNet). In 2019 IEEE Automatic Speech Recognition and Understanding Workshop (ASRU), pp. 667–673, 2019.
> > > >
> > > > [3] Ruohan Gao, Kristen Grauman. Visualvoice: Audio-visual speech separation with crossmodal consistency. IEEE/CVF Conference on Computer Vision and Pattern Recognition (CVPR), pp. 15490-15500. IEEE, 2021
> > > >
> > > > [4] Hector Martel, Julius Richter, Kai Li, Xiaolin Hu, et al. Audio-visual speech separation in noisy environments with a lightweight iterative model. In Conference of the International Speech Communication Association (ICASSP). ISCA, 2023.
> > > >
> > > > [5] Samuel Pegg, Kai Li, and Xiaolin Hu. RTFS-Net: Recurrent time-frequency modelling for efficient audio-visual speech separation. In The Twelfth International Conference on Learning Representations, 2023.
> > > >
> > > > [6] Kai Li, Runxuan Yang, et al. IIANet: an intra- and inter-modality attention network for audio-visual speech separation. In Proceedings of the 41st International Conference on Machine Learning, pp. 29181–29200, 2024b.

---

### Official Review · Reviewer_1MMg · 2025-11-01

**Soundness:** 3
**Presentation:** 3
**Contribution:** 3
**Rating:** 4
**Confidence:** 4

**Summary:**

This paper presents an audio-visual model for speech separation. The main contribution is a model which achieves SOTA results and is also lightweight. Results on multiple datasets and an ablation study are presented.

**Strengths:**

- The presentation of an audio-visual speech seperation model which achieves SOTA performance and is also lightweights is a useful contribution.
- Results on multple datasets and detailed ablation study.

**Weaknesses:**

- The title is a bit misleading. The model recovers the speech of the main speaker only, and not all speech signals at the same time. So speech enhancement might have been a better term for this task (and it's more consistent with the literature).
- The paper does not compare with some recent SOTA models, e..g. "LA-VocE: Low-SNR audio-visual speech enhancement using neural vocoders
- It's not clear what results (in terms of number of background speakers) are presented in Section 5. Given that the appendix includes results with multiple background speakers, then these results probably include 1 background speaker. Also, why not testing the model's performance on different types of background noise and combination of background noise+multiple speakers.
- No supplementary material is provided, so it's hard to judge how good the denoised samples are. The numbers look good, but without listening to examples it's hard to judge the quality. Some examples are provided on the provided link, but they are too few. There are only 6 examples where the model is compared to other models. Would also be good if some details about the type of noise in each example are added.

**Questions:**

Please see above.

---

> ### Author Response · Authors · 2025-11-20
>
> **Q1: The title is a bit misleading. The model recovers the speech of the main speaker only, and not all speech signals at the same time. So speech enhancement might have been a better term for this task (and it's more consistent with the literature).**
>
> **A1:** We thank the reviewer for the thoughtful suggestion. We acknowledge the concern regarding the distinction between "separation," "extraction," and "enhancement".
>
> However, we respectfully suggest that "Speech Separation" remains an appropriate and accurate descriptor for our work for two main reasons:
>
> - Nature of Interference: In the signal processing community, "Speech Enhancement" typically implies removing non-speech background noise (denoising). In contrast, our task focuses on disentangling a speech signal from a mixture of competing speech signals (the cocktail party problem). Since the primary challenge is separating one voice from another, rather than just suppressing noise, "Separation" is the more distinct and widely used term for this problem setting.
>
> - Consistency with Literature: As noted in the query, the term "Speech Separation" is widely adopted in the audio-visual community to describe the task of extracting the speech of the visually corresponding speaker from a mixture. Prior works such as [1, 2, 3, 4, 5, 6, 7, 8] use this terminology even when the model output is a single stream.
>
> Therefore, we believe keeping "Speech Separation" in the title accurately reflects the problem of disentangling overlapping speech. However, to be precise, we can clarify this as "Target Speech Separation" in the main text to avoid ambiguity.
>
> **Q2: The paper does not compare with some recent SOTA models, e..g. "LA-VocE: Low-SNR audio-visual speech enhancement using neural vocoders".**
>
> **A2:** We thank the reviewer for this suggestion. Regarding comparisons with recent state-of-the-art models, we would like to clarify the following: our experiments already include several of the most recently published methods, such as IIANet (ICML 2024) [1], SwiftNet (ECAI 2025) [2], and AV-Mossformer2 (Interspeech 2025) [10]. Among these baselines, AV-Mossformer2 currently achieves the strongest reported performance on the VoxCeleb2 dataset.
>
> With respect to the reviewer’s example, LA-VocE (ICASSP 2023), we conducted a detailed investigation. The method is not open-sourced, and the original paper does not evaluate on the standard AVSS benchmark datasets commonly used in the field (LRS2, LRS3, VoxCeleb2), which makes a direct comparison difficult. Nevertheless, we attempted to reproduce the method and evaluated it on the LRS2 dataset. It is important to note that LA-VocE employs HiFiGAN as its vocoder, whose output is not temporally aligned with the ground-truth waveform, making standard metrics such as SI-SNR inapplicable. Therefore, we report comparisons only using PESQ.
>
> | Methods | PESQ |
> | :--- | :---: |
> | LA-VocE | 2.50 |
> | **Dolphin** | **3.29** |
>
> The results show that our method still achieved superior speech quality compared to LA-VocE.

---

> ### Author Response · Authors · 2025-11-20
>
> **Q3: It's not clear what results (in terms of number of background speakers) are presented in Section 5. Given that the appendix includes results with multiple background speakers, then these results probably include 1 background speaker. Also, why not testing the model's performance on different types of background noise and combination of background noise+multiple speakers.**
>
> **A3:** Regarding the number of speakers in Section 5, we would like to clarify the following: as stated in Section 4 (Dataset), all mixtures used in our experiments consist of two-speaker mixtures. Therefore, the results correspond to scenarios with one interfering (background) speaker. We further note that our model operates by separating all distinct speakers present in the input mixture.
>
> As for why we did not initially evaluate under additional types of background noise, our primary intention was to follow the established evaluation protocol in the AVSS literature to ensure a fair and direct comparison with prior work [1,2,3,4,5,7]. These studies predominantly evaluate on standard two-speaker mixtures constructed from LRS2, LRS3, and VoxCeleb2. Importantly, these datasets (e.g., LRS2, VoxCeleb2) originate from real-world sources (e.g., BBC broadcasts, YouTube videos), and thus their mixtures naturally contain certain levels of environmental noise.
>
> Nonetheless, we fully agree with the reviewer that assessing robustness under more diverse acoustic conditions is crucial. Following your suggestion, we conducted an additional set of experiments. Using VoxCeleb2 as the base, we introduced environmental noise (from FSD50K [9]) and music noise (from FMA [10]), and constructed four more challenging test scenarios. Each scenario contains 1000 test samples:
>
> - **Scenario 1 (Environmental Noise)**: 1 speaker + environmental noise
> - **Scenario 2 (Music Noise)**: 1 speaker + music noise
> - **Scenario 3 (Environmental Noise + 2 Speakers)**: 1 speaker + environmental noise + 2 additional interfering speakers (3 speakers total)
> - **Scenario 4 (Music Noise + 2 Speakers)**: 1 speaker + music noise + 2 additional interfering speakers (3 speakers total)
>
> We evaluated our model (Dolphin) as well as two representative baselines (IIANet [1] and AV-Mossformer2 [11]) across all scenarios. The results, summarized in the appendix, show that Dolphin consistently outperforms the baselines in all metrics (SI-SNRi, SDRi, PESQ) across all four challenging conditions. Notably, under the most adverse cases involving three speakers plus environmental or music noise (Scenarios 3 and 4), Dolphin demonstrates substantially greater robustness and separation capability.
>
> We sincerely appreciate the reviewer’s suggestion. These additional experiments significantly strengthen the empirical evidence supporting our method, and we have included the full results in the revised manuscript appendix.
>
> > Scenario 1 (Environmental Noise): 1 speaker + environmental noise
>
> | Methods | SI-SNRi ↑ | SDRi ↑ | PESQ ↑ |
> | :--- | :---: | :---: | :---: |
> | IIANet | 6.69 | 9.71 | 2.07 |
> | AV-Mossformer2 | 8.33 | 9.01 | 2.13 |
> | **Dolphin (ours)** | **11.41** | **12.36** | **2.43** |
>
> > Scenario 2 (Music Noise): 1 speaker + music noise
>
> | Methods | SI-SNRi ↑ | SDRi ↑ | PESQ ↑ |
> | :--- | :---: | :---: | :---: |
> | IIANet | 2.04 | 5.92 | 1.64 |
> | AV-Mossformer2 | 6.68 | 7.53 | 1.76 |
> | **Dolphin (ours)** | **7.49** | **8.52** | **1.85** |
>
> > Scenario 3 (Environmental Noise + 2 Speakers): 1 speaker + environmental noise + 2 additional interfering speakers (3 speakers total)
>
> | Methods | SI-SNRi ↑ | SDRi ↑ | PESQ ↑ |
> | :--- | :---: | :---: | :---: |
> | IIANet | 1.09 | 4.64 | 1.44 |
> | AV-Mossformer2 | 3.33 | 4.71 | 1.49 |
> | **Dolphin (ours)** | **5.37** | **6.56** | **1.57** |
>
> > Scenario 4 (Music Noise + 2 Speakers): 1 speaker + music noise + 2 additional interfering speakers (3 speakers total)
>
> | Methods | SI-SNRi ↑ | SDRi ↑ | PESQ ↑ |
> | :--- | :---: | :---: | :---: |
> | IIANet | 0.54 | 3.63 | 1.36 |
> | AV-Mossformer2 | 3.90 | 4.56 | 1.41 |
> | **Dolphin (ours)** | **4.11** | **5.33** | **1.46** |

---

> > ### Author Response · Authors · 2025-11-20
> >
> > **Q4: No supplementary material is provided, so it's hard to judge how good the denoised samples are. The numbers look good, but without listening to examples it's hard to judge the quality. Some examples are provided on the provided link, but they are too few. There are only 6 examples where the model is compared to other models. Would also be good if some details about the type of noise in each example are added.**
> >
> > **A4:** To clarify, we did provide audio demonstrations via the anonymous link included in our initial submission. However, we acknowledge your concern regarding the limited number of initial examples and have significantly expanded the demo page contents to facilitate a comprehensive evaluation. Specifically, we have increased the number of test cases from the original 6 to 150 distinct samples. These samples are now systematically organized into five specific acoustic scenarios to detail the noise types: 0 Noise + 1 Speaker, 1 Music + 0 Speaker, 1 Music + 2 Speakers, 1 Noise + 0 Speaker, and 1 Noise + 2 Speakers. Each category contains 30 samples, providing a rigorous auditory comparison between our Dolphin model and representative baselines (IIANet [1] and AV-Mossformer2 [11]) across diverse and challenging interference conditions.
> >
> > ### Reference
> > [1] Kai Li, Runxuan Yang, et al. IIANet: an intra- and inter-modality attention network for audio-visual speech separation. In Proceedings of the 41st International Conference on Machine Learning, pp. 29181–29200, 2024b.
> >
> > [2] Wendi Sang, Kai Li, et al. A Fast and Lightweight Model for Causal Audio-Visual Speech Separation. arXiv preprint, 2025.
> >
> > [3] Ariel Ephrat, Inbar Mosseri, et al. Looking to Listen at the Cocktail Party: A Speaker-Independent Audio-Visual Model for Speech Separation. In Proceedings of the IEEE/CVF Conference on Computer Vision and Pattern Recognition (CVPR), 2018.
> >
> > [4] Soo-Whan Chung, Soyeon Choe, et al. FaceFilter: Audio-Visual Speech Separation Using Still Images. arXiv preprint, 2020.
> >
> > [5] Lee, Suyeon, et al. Seeing through the conversation: Audio-visual speech separation based on diffusion model. IEEE International Conference on Acoustics, Speech and Signal Processing (ICASSP), 2024.
> >
> > [6] Pan, Tianrui, et al. RAVSS: Robust Audio-Visual Speech Separation in Multi-Speaker Scenarios with Missing Visual Cues. Proceedings of the 32nd ACM International Conference on Multimedia (MM), 2024.
> >
> > [7] Martel, Héctor, et al. Audio-Visual Speech Separation in Noisy Environments with a Lightweight Iterative Model. INTERSPEECH, 2023.
> >
> > [8] Lu, Rui, Zhiyao Duan, and Changshui Zhang. Audio–visual deep clustering for speech separation. IEEE/ACM Transactions on Audio, Speech, and Language Processing, 2016.
> >
> > [9] Eduardo Fonseca, Xavier Favory, et al. FSD50K: An Open Dataset of Human-Labeled Sound Events. In Proceedings of the IEEE International Conference on Acoustics, Speech and Signal Processing (ICASSP), pp. 776–780, 2022.
> >
> > [10] Michaël Defferrard, Kirell Benzi, et al. FMA: A Dataset for Music Analysis. In Proceedings of the 18th International Society for Music Information Retrieval Conference (ISMIR), pp. 316–323, 2017.
> >
> > [11] Shengkui Zhao, Zexu Pan, et al. ClearerVoice-Studio: Bridging Advanced Speech Processing Research and Practical Deployment. In Interspeech 2025, 5 pages, 2025.

---

### Official Review · Reviewer_Pm3b · 2025-11-03

**Soundness:** 3
**Presentation:** 3
**Contribution:** 3
**Rating:** 6
**Confidence:** 3

**Summary:**

This paper tackles audio-visual speech separation (AVSS). The core problem it targets is that current SOTA methods either achieve good separation quality but rely on a heavy visual backbone plus multiple iterations of the separator, or they are lightweight but require repeated iterations, which hurts real-time performance and deployability. The authors propose an overall framework called Dolphin, with two main components: a lightweight dual-path video encoder DP-LipCoder and a single-iteration encoder–decoder separator with Global–Local Attention.

**Strengths:**

It makes explicit a long-standing but very real issue in AVSS that the visual encoder is too heavy, and then deliberately designs a dual-path + VQ + distillation structure on the visual side. This is a more thoughtful route than simply plugging in an off-the-shelf lipreading backbone or adding a small autoencoder. It jointly considers both reconstructive and semantic aspects, rather than just trimming the backbone.

The separator also states the efficiency goal of replacing multiple iterations with a single pass quite clearly, and then compensates for it with global and local components.

**Weaknesses:**

The paper clearly states that compressing lip-reading backbones causes semantic loss, and that purely reconstruction-oriented lightweight encoders only capture shallow, pixel-level cues. It then proposes a dual-path design with VQ-based discrete semantics plus distillation to address this. However, the mechanism of why the dual-path + discrete tokens specifically preserve the task-relevant semantics under heavy compression is mostly justified empirically (Table 1), rather than analytically. A more explicit explanation or diagnostic would strengthen the claim.

**Questions:**

The proposed GLA-based separator is presented as a single-pass design for efficiency. However, since the block itself is a generic feature transform, it seems technically feasible to unroll it for multiple passes (as commonly done in audio-visual separation) to trade extra compute for further gains. I would like to see how the model performs under multiple iterations.

---

> ### Author Response · Authors · 2025-11-20
>
> **Q1: The mechanism of why the dual-path + discrete tokens specifically preserve the task-relevant semantics under heavy compression is mostly justified empirically (Table 1), rather than analytically. A more explicit explanation or diagnostic would strengthen the claim.**
>
> **A1:** Our dual-path design is motivated by the observation that a lightweight video encoder trained solely with a reconstruction objective tends to overfit “shortcut cues” such as pixel-level appearance and local textures, making it difficult to learn semantics that are aligned with speech. On the other hand, aggressively compressing a large lip-reading backbone substantially weakens its ability to preserve audio-aligned semantic information. Our goal is to obtain a highly compressed yet semantically sufficient and minimally redundant visual representation, while still retaining the low-level structural cues (e.g., lip geometry) that remain important for the separation task.
>
> Implementing a single-path architecture is suboptimal because forcing the semantic encoder to handling reconstruction inevitably compels it to capture fine-grained pixel details. This contaminates the codebook with irrelevant variations and dilutes semantic density. To avoid this, we adopt a dual-path design to decouple these objectives: the semantic path is trained solely with VQ and distillation losses to ensure semantic purity, while the reconstruction path independently preserves the geometric structure required for separation.
>
> Regarding the use of discrete tokens, we leverage Vector Quantization (VQ) as an information bottleneck to distill semantics. The objective is to maximize the mutual information between the latent representation $Z$ and the audio signal ($I(Z;\text{Audio})$), while discarding irrelevant visual noise. By enforcing a finite-capacity codebook, VQ converts $Z$ into a discrete variable, which naturally constrains its entropy ($H(Z) \le \log|\text{Codebook}|$). This constraint compels the encoder to filter out fine-grained, pixel-level variations that are unrelated to speech, retaining only the structural patterns necessary for the task. Furthermore, the distillation process explicitly steers these discrete tokens toward the AV-HuBERT audio-aligned semantic space. The combination of this bottleneck effect and semantic guidance yields a compact, minimal sufficient representation that serves as a robust prior for the separator.
>
> We have added a detailed discussion of this rationale in the revised manuscript.
>
> **Q2: how the model performs under multiple iterations.**
>
> **A2:** We thank the reviewer for the constructive suggestion. We agree that the GLA-based separator is a generic feature transformation module, and thus it is technically feasible to unroll it for multiple passes, similar to prior AVSS approaches, in order to gain additional performance at the cost of increased computation. To verify this, we extended our default single-pass configuration to two and four iterations while keeping all other components unchanged. The results are shown in the table below.
>
> | Iterations | SI-SNRi ↑ | SDRi ↑ | PESQ ↑ | Params (M) | MACs (G) |
> | :---: | :---: | :---: | :---: | :---: | :---: |
> | 1 | 16.8 | 16.9 | 3.29 | 6.22 | 8.51 |
> | 2 | 17.0 | 17.2 | 3.29 | 6.22 | 18.44 |
> | 4 | **17.5** | **17.6** | **3.31** | 6.22 | 34.05 |
>
> We observed that multiple iterations yield small but consistent improvements (approximately +0.7 dB SI-SNRi from 1 to 4 iterations), but the computational cost increases multiplicatively (around 4× MACs). This indicates that our single-pass GLA separator already captures the majority of the gains typically provided by iterative refinement, achieving a more favorable balance between performance and efficiency.

---

### Official Review · Reviewer_Qs5W · 2025-11-10

**Soundness:** 3
**Presentation:** 4
**Contribution:** 3
**Rating:** 6
**Confidence:** 3

**Summary:**

The paper presents a more computationally efficient AV speech separation system called Dolphin. As part of this system, the author's also present a lightweight visual encoder that maps video frames to semantic, well-aligned-to-audio discrete tokens. Together they yield state-of-the-art quality across a variety of corpora and metrics while being fairly computationally efficient.

**Strengths:**

The paper demonstrates an approach for building a state-of-the-art speech separation system. They modify TDANet, moving from iterative, progressive separation to single separation step for efficiency with joint local and global attention for each layer for a large win on efficiency. They also note the need for improving throughput in the vision encoder, which encodes lip motion frames into discrete semantic tokens, by developing a fast, high quality model called DP-LipCoder. Third, for separation they use an efficient, mixed global/local attention encoder/decoder. These innovations yield a system that surpasses many recent AV speech separation systems.

Further, the system is efficient showing less memory use, fewer MACs, faster GPU inference than many competing systems.

The paper will provide a github repo that will allow others to fully reproduce results as much as possible.

**Weaknesses:**

In some ways, this paper is an incremental approach to AVSS, not providing many new insights, however strong the results are. Critics may wonder if human evaluations/side-by-sides would agree that the Dolphin system is indeed superior to the other approaches in terms of quality improvement and fewer artifacts. Lastly, the tasks all appear to be artificially created speaker overlap and equation (1) is a large simplication--with the Lombard effect the noise causes non-linear changes to speech production.

**Questions:**

How well does the Dolphin system do with real overlapping speakers?
Have conducted any human listening tests / comparative evals against other systems?

---

> ### Author Response · Authors · 2025-11-20
>
> **Q1: In some ways, this paper is an incremental approach to AVSS, not providing many new insights, however strong the results are.**
>
> **A1:** We thank the reviewer for this critical observation. We would like to clarify that we identified a fundamental conflict in designing the video encoder for AVSS and proposed Dolphin as an good solution. Specifically, we provide new insights in two dimensions: (1) resolving the semantics-reconstruction conflict in visual encoding, and (2) overcoming the latency-performance trade-off in separation.
>
> 1. New Insight in Visual Encoding: The Necessity of Decoupling (Dual-Path)
>
> To answer the question of "new insights," we invite the reviewer to consider the two conventional paradigms for lightweight visual encoding and why they fail to capture the necessary semantics:
>
> - **Path A (Reconstruction-driven):** Standard lightweight encoders trained via video reconstruction often succumb to "shortcut learning." They overfit to low-level pixel cues (lighting, texture) to satisfy reconstruction loss, failing to capture the high-level lip motion semantics required for separation.
> - **Path B (Compression):** Conversely, aggressively pruning heavy backbones (e.g., 3D ResNet-18 [1,2]) destroys fragile, audio-aligned semantic information.
>
> Our Core Insight: We identify that semantic extraction and visual reconstruction are conflicting objectives in a lightweight constrained setting. If we force a single semantic pathway to also handle reconstruction, it is compelled to encode fine-grained pixel details, which "pollutes" the codebook with task-irrelevant variance and lowers semantic density.
>
> Our Solution (DP-LipCoder): We introduce a dual-path architecture to explicitly decouple these goals.
>
> - **Separation of Concerns:** We route all reconstruction supervision to a dedicated path (preserving geometry/structure), while the semantic path is trained exclusively under Vector Quantization (VQ) and Distillation constraints.
>
> - **Information Bottleneck:** This design maximizes $I(Z; \text{Audio})$ while minimizing $I(Z; \text{VideoNoise})$. The VQ mechanism acts as an entropy constraint ($H(Z) \le \log|\text{Codebook}|$) to suppress fine-grained noise, while distillation aligns the representation with the AV-HuBERT space [3]. This creates a "Minimal Sufficient Representation" that provides a robust prior for the separator.
>
> 2. Efficiency Insight in Separation: Single-Iteration Multi-Scale Attention (GLA)
>
> Prior efficient AVSS models often rely on iterative refinement (e.g., RTFSNet [4] or AVLiT [5]), which reduces parameter counts but introduces latency due to repeated processing steps.
>
> Our Solution: We introduce an efficient single-iteration separator with multi-scale global–local attention (GLA). Instead of iterative loops, our encoder–decoder separator performs separation in a single forward pass. We incorporate GLA blocks at each layer to capture multi-scale dependencies: Global Attention (GA) models long-range context, while Local Attention (LA) adopts heat diffusion attention (HDA) to preserve fine details and suppress noise.
>
> Results: This design yields SOTA performance with >50% fewer parameters, 2.4× lower MACs, and up to 6× faster GPU inference compared to IIANet [6] (Tables 3 & 4, Sec. 5.2).
>
> We believe these two directions: (i) the information-theoretic decoupling of lip semantics from pixel reconstruction, and (ii) the non-iterative global-local collaborative separation, provide verifiable new insights that go beyond incremental engineering. They offer a fresh perspective on balancing semantic density with computational efficiency.

---

> ### Author Response · Authors · 2025-11-20
>
> **Q2: How does the Dolphin system perform in real overlapping speaker scenarios? Has any subjective listening test or comparative evaluation with other systems been conducted?**
>
> **A2:** First, we would like to clarify that the standard AVSS datasets commonly used in prior work (LRS2, LRS3, and VoxCeleb2) are indeed artificially constructed overlapping-speaker scenarios. This setup is not specific to our method; rather, it is a community-wide practice that enables supervised training and standardized evaluation.[1,2,3,4,5,6,7,8] We also note that our demo page already includes several examples of real overlapping speech mixtures recorded in natural environments, in which speakers naturally adjust their vocal effort in noisy conditions, a setting that inherently reflects the Lombard effect and therefore differs substantially from artificially mixed data.
>
> We fully agree that evaluating systems on natural, in-the-wild speech with human listeners is crucial for demonstrating practical effectiveness. To this end, we conducted a Mean Opinion Score (MOS) study using challenging real-world overlapping speech data. We searched YouTube debate videos and collected 18 clips (10 to 20 seconds each). These recordings contain spontaneous overlapping speech, background noise, and room reverberation, which are conditions that differ substantially from artificially mixed datasets.
>
> We recruited 50 human listeners to rate the separation results. Each listener evaluated the audio produced by IIANet, AV-Mossformer2, and our Dolphin system on a 5-point scale (1 = poor, 5 = excellent). The listeners were blind to the model identities, and the playback order was randomized.
>
> The average MOS results are summarized in the table below. The proposed Dolphin system achieved significantly higher scores across all three metrics, particularly in separation quality and audio fidelity. Listeners consistently judged Dolphin’s outputs as clearer and containing substantially fewer artifacts compared to competing methods. These quantitative findings align with the qualitative examples shown in our demo page, confirming that human evaluators indeed perceive Dolphin as producing higher-quality, lower-artifact separations in real-world settings. We have added this study to the appendix of the revised manuscript.
>
> | Methods | MOS |
> | :--- | :---: |
> | IIANet | 2.24 ± 0.15 |
> | AV-Mossformer2 | 2.85 ± 0.27 |
> | **Dolphin (ours)** | **3.86 ± 0.25** |
>
> ### Reference
>
> [1] Ruohan Gao, Kristen Grauman. Visualvoice: Audio-visual speech separation with crossmodal consistency. IEEE/CVF Conference on Computer Vision and Pattern Recognition (CVPR), pp. 15490-15500. IEEE, 2021.
>
> [2] Jian Wu, Yong Xu, et al. Time domain audio visual speech separation. In IEEE automatic speech recognition and understanding workshop (ASRU), pp. 667–673. IEEE, 2019.
>
> [3] Bowen Shi, Wei-Ning Hsu, et al. Learning audio-visual speech representation by masked multimodal cluster prediction. In International Conference on Learning Representations, 2022.
>
> [4] Samuel Pegg, Kai Li, and Xiaolin Hu. RTFS-Net: Recurrent time-frequency modelling for efficient audio-visual speech separation. In The Twelfth International Conference on Learning Representations, 2023.
>
> [5] Hector Martel, Julius Richter, Kai Li, Xiaolin Hu, et al. Audio-visual speech separation in noisy environments with a lightweight iterative model. In Conference of the International Speech Communication Association (ICASSP). ISCA, 2023.
>
> [6] Kai Li, Runxuan Yang, et al. IIANet: an intra- and inter-modality attention network for audio-visual speech separation. In Proceedings of the 41st International Conference on Machine Learning, pp. 29181–29200, 2024b.
>
> [7] Suyeon Lee, Chaeyoung Jung, et al. Seeing Through the Conversation: Audio-Visual Speech Separation based on Diffusion Model. arXiv preprint arXiv: 2310.19581, 2023.
>
> [8] Tianrui Pan, Jie Liu, et al. RAVSS: Robust Audio-Visual Speech Separation in Multi-Speaker Scenarios with Missing Visual Cues. arXiv preprint arXiv: 2407.19224, 2024.

---

### Author Response · Authors · 2025-11-26
**Hope the reviewers will take note of our response**

Dear Reviewers,

After submitting the initial comments, we incorporated your feedback into a revised version of our paper, performed some additional experiments as you requested, and wrote a response to address your main concerns.

We hope to interact with you during the discussion and potentially further improve the quality of our paper.

Thank you very much in advance.

Kind regards,

Authors

---

### Meta-Review · Area_Chair_DTqH · 2025-12-12

**Summary:**

Three out of four reviewers are positive with respect to the paper and contributions. The fourth reviewer is "4: marginally below the acceptance threshold. But would not mind if paper is accepted"

The authors provided responses to most of the expressed concerns.

The ACs carefully read the paper, the reviews, and the authors' responses.
The ACs agree with the reviewers that the paper makes significant contributions and can be accepted. The authors are invited to benefit from the feedback received and further improve their camera ready paper.

**Reviewer Concerns:**

Three out of four reviewers are positive with respect to the paper and contributions. The fourth reviewer is "4: marginally below the acceptance threshold. But would not mind if paper is accepted"

The authors provided responses to most of the expressed concerns.

The ACs carefully read the paper, the reviews, and the authors' responses.
The ACs agree with the reviewers that the paper makes significant contributions and can be accepted. The authors are invited to benefit from the feedback received and further improve their camera ready paper.

**Reviewer Scores:**

Three out of four reviewers are positive with respect to the paper and contributions. The fourth reviewer is "4: marginally below the acceptance threshold. But would not mind if paper is accepted"

The authors provided responses to most of the expressed concerns.

The ACs carefully read the paper, the reviews, and the authors' responses.
The ACs agree with the reviewers that the paper makes significant contributions and can be accepted. The authors are invited to benefit from the feedback received and further improve their camera ready paper.

---

### Decision · Program_Chairs · 2026-01-26

Accept (Poster)